# Understanding variability in petroleum jet fuel life cycle greenhouse gas emissions to inform aviation decarbonization

Liang Jing [1,2], Hassan M. El-Houjeiri[3], Jean-Christophe Monfort [3], James Littlefield [2], Amjaad Al-Qahtani[3], Yash Dixit [4], Raymond L. Speth [4], Adam R. Brandt [5], Mohammad S. Masnadi [6], Heather L. MacLean[7], William Peltier[8], Deborah Gordon [9] & Joule A. Bergerson [1] ✉

A pressing challenge facing the aviation industry is to aggressively reduce greenhouse gas emissions in the face of increasing demand for aviation fuels. Climate goals such as carbon-neutral growth from 2020 onwards require continuous improvements in technology, operations, infrastructure, and most importantly, reductions in aviation fuel life cycle emissions. The Carbon Offsetting Scheme for International Aviation of the International Civil Aviation Organization provides a global market-based measure to group all possible emissions reduction measures into a joint program. Using a bottom-up, engineering-based modeling approach, this study provides the first estimates of life cycle greenhouse gas emissions from petroleum jet fuel on regional and global scales. Here we show that not all petroleum jet fuels are the same as the country-level life cycle emissions of petroleum jet fuels range from 81.1 to 94.8 $gCO_2e$ $MJ^{-1}$, with a global volume-weighted average of 88.7 $gCO_2e$ $MJ^{-1}$. These findings provide a high-resolution baseline against which sustainable aviation fuel and other emissions reduction opportunities can be prioritized to achieve greater emissions reductions faster.

The global aviation industry is a facilitator of many economic activities and a fast-growing economic contributor, providing more than 80 million jobs and accounting for 3.5% of global GDP[1]. The International Civil Aviation Organization (ICAO) has estimated that petroleum jet fuel (hereafter referred to as "jet fuel") consumption could be 2.2-3.1 times higher than its 2015 level by 2045 (based on a pre-COVID-19 annual growth rate of 4%)[2]. However, this global connectivity of people and goods has significant climate impacts. Civil aviation via jet fuel combustion accounts for 2% of global anthropogenic greenhouse gas

(GHG) emissions[3]. Although aviation is not directly included in the Paris Agreement, the aviation sector set three goals in 2009 to address their GHG emissions: improving fuel efficiency by 1.5% annually; implementing carbon-neutral growth from 2020 onwards; and cutting 2005 emissions levels in half by 2050[4]. Although the fuel efficiency target has been constantly met with a 2.4% average annual improvement via advancements in technology, operations, and infrastructure, the remaining two goals have yet to be achieved. In addition, the IPCC AR6 report notes that a rapid and far-reaching energy transition is

[1]Department of Chemical and Petroleum Engineering, Schulich School of Engineering, University of Calgary, Calgary, Alberta, Canada. [2]Climate and Sustainability Group, Aramco Research Center–Detroit, Aramco Americas, Novi, MI, USA. [3]Energy Traceability Technology, Technology Strategy and Planning, Saudi Aramco, Dhahran, Saudi Arabia. [4]Laboratory for Aviation and the Environment, Department of Aeronautics and Astronautics, Massachusetts Institute of Technology, Cambridge, MA, USA. [5]Department of Energy Resources Engineering, School of Earth, Energy & Environmental Sciences, Stanford University, Stanford, CA, USA. [6]Chemical and Petroleum Engineering Department, University of Pittsburgh, Pittsburgh, PA, USA. [7]Department of Civil and Mineral Engineering; Department of Chemical Engineering and Applied Chemistry, University of Toronto, Toronto, ON, Canada. [8]Downstream Advisors, Inc., Dallas, TX, USA. [9]Watson Institute for International and Public Affairs, Brown University, Providence, RI, USA and RMI, Boulder, CO, USA. ✉e-mail: jbergers@ucalgary.ca

required to ensure global warming is held to 1.5–2 °C[1,4,5]. In response to mounting climate concerns, the Carbon Offsetting Scheme for International Aviation (CORSIA) was adopted as a voluntary carbon offset and emissions reduction program in 2016 and has been applied to international aviation since 2019. As a global market-based measure, CORSIA can assist the ICAO's goal of carbon-neutral growth via CORSIA-eligible fuels (CEFs) to reduce the industry's emissions offsetting requirements[6].

CEFs can be classified into sustainable aviation fuel (SAF) and lower carbon aviation fuel (LCAF). SAF is produced from biomass or waste-derived sources such as biofuels or via power-to-liquid processes (i.e., low-carbon synthetic fuels produced from $CO_2$ and renewable hydrogen via electrolysis). It meets the CORSIA sustainability criteria and can be directly used as drop-in fuel without engine or system modifications. ICAO has advanced the development of a process-based attributional life cycle assessment (LCA) approach to calculate the baseline life cycle GHG emissions of jet fuel and GHG emissions of biomass-based SAF throughout the supply chain[6,7]. The results can be used to calculate emissions reductions from SAF against the conventional jet fuel life cycle GHG emissions baseline of 89.0 $gCO_2e$ $MJ^{-1}$ as adopted in Annex 16 Vol IV to the Convention of International Civil Aviation[8]. Biomass-based and power-to-liquid SAF can cut up to 80-90% of petroleum baseline emissions, however, concerns have been raised about high production cost and low scalability. There are additional potential constraints on biomass-based SAF, such as large variability in emissions reduction potential, supply shortages, land-use change impacts, and the demand for fertilizers and pesticides to produce biomass feedstock[1,3,9,10]. LCAF is a petroleum-based jet fuel with at least 10% life cycle GHG emissions reduction from the same baseline (89.0 $gCO_2e$ $MJ^{-1}$)[8]. Such reductions can be achieved–for example, through carbon capture and storage (CCS) and renewable energy utilization in the oil supply chain[11]. LCAF was approved as CORSIA Eligible Fuel in Annex 16 Vol IV[8], and the development of LCA methods to assess its climate benefits is ongoing.

In meeting the emissions reduction targets set by ICAO and International Air Transport Association (IATA), and informing regional low-carbon jet fuel policies and measures, there is an increasing need to transparently demonstrate baseline life cycle GHG emissions and potential reductions of jet fuel at both global and regional scales. There have been numerous life cycle carbon footprint studies on SAF[12–16] but the understanding of LCAF has been limited due to the lack of geographically rich data (e.g., crude intake and jet fuel yield by refinery) and inconsistent assumptions (e.g., system boundary and allocation method) that result in a wide range of life cycle emissions estimates varying up to five-fold[17–26]. To fill these gaps, we perform a first-of-its-kind LCA that estimates the life cycle (i.e., well-to-wake, which describes the life cycle inventory) GHG emissions of petroleum jet fuel (rather than SAF) on a global scale. By applying engineering-based tools (i.e., OPGEE[27], COPTEM[28], and PRELIM[29], see Methods) using public and commercial datasets[30,31], we provide traceability of the carbon footprint of each jet fuel pathway from the market to its source oil field. More importantly, such a high-resolution baseline can help the aviation industry prioritize SAF and other emissions reduction opportunities to achieve climate goals faster and more effectively.

## Results and Discussion
### Global jet fuel market
Global jet fuel consumption was ~6.7 million barrels per day (Mbbl $d^{-1}$) in 2017, according to the International Energy Agency (IEA) World Oil Statistics[32]. Most countries produce the jet fuel they consume. On average, domestic production represents 81% of the jet fuel consumed in a country[32,33]. A country's jet fuel consumption is highly dependent on its GDP, with a Pearson correlation coefficient $R$ equal to 0.98. The US and China are the largest consumers of jet fuel, and they both have a disproportionately high number of deep conversion refineries[34], which are

geared to make more light and middle distillates. Among the Organisation for Economic Co-operation and Development (OECD) countries, the US, Japan, and South Korea have the highest potential to be self-sufficient in jet fuel refining, with domestic production representing over 90% of their demand. The noticeable exceptions are France (58%), Germany (54%), Italy (53%), the UK (43%), and Australia (40%).

The global jet fuel trade was ~2 Mbbl $d^{-1}$ in 2017. In order of export and import volumes, South Korea, the US, Netherlands, Singapore, India, China, and UAE are leading exporting countries. At the same time, the UK, the US, Germany, Australia, and France are the top importing countries. As the world's largest jet fuel exporter, South Korea's exports represent 70% of its domestic production, making up 16% of global exports and 58%, 86%, and 48% of the US, Japan, and Australia's imports, respectively. Major European OECD countries, including the UK, Germany, France, and Italy, on the other hand, rely heavily on imports that meet 50-70% of their demand (Fig. 1b). This lack of self-sufficiency in jet fuel may be due to refining capacity constraints, market demand and pricing for refined products, and jet fuel price differential across the globe[34]. In 2017, the refinery throughput and petroleum product demand in the EU were 686 and 762 million tonnes, respectively, indicating a small gap between local production and consumption[35]. However, this small gap is widening as EU refining volumes decline. It also hides the increasing imbalance between the demand and supply of different refining products. For example, EU relies heavily on diesel/jet fuel imports from Russia, the U.S., and the Middle East while gasoline overproduction has caused uneconomic conditions in many refineries in EU[35,36]. Future work is needed to understand better the implications of decarbonizing the oil and gas sector while considering the imbalances created by reducing individual petroleum products on different time scales.

### Well-to-wake life cycle carbon intensity (CI) of jet fuel
Supplementary Fig. 1 shows the LCA methods and system boundary defined in this study. The global volume-weighted average jet fuel well-to-wake carbon intensity (CI) estimate is close to the ICAO conventional jet fuel baseline[3,6,37] (88.7 as compared to 89.0 $gCO_2e$ $MJ^{-1}$, respectively). This translates to annual emissions of ~1.2 $GtCO_2e$ per year from global aviation (based on ~6.7 Mbbl per day consumption), equivalent to 3.6% of the global energy-related carbon dioxide emissions in 2017[38].

As shown in Supplementary Fig. 2, the country-level well-to-wake CI ranges from 81.1 to 94.8 $gCO_2e$ $MJ^{-1}$ (also see Supplementary Table 1), mainly due to variations in crude oil production emissions. Turkey has the highest CI among major jet fuel-consuming countries because 97% of its jet fuel consumption is from domestic refineries, which are dominated by crude oil imported from Iran and Iraq that has higher crude oil production CI (12.9-21.4 $gCO_2e$ $MJ^{-1}$ jet fuel) than the global average. A similar trend is observed in Iran. For each MJ of jet fuel produced by its refineries, 14.2 $gCO_2e$ emissions are emitted from crude oil production. Australia and Canada are also among the high CI countries, primarily because they import a significant amount (i.e., 100 and 50 kbbl $d^{-1}$, respectively, equivalent to 67% and 39% of their consumption volume) of jet fuel through long-distance transport to end-users (their distribution CI are 5.2 and 2.9 $gCO_2e$ $MJ^{-1}$, respectively, higher than the global volume-weighted average of 0.8 $gCO_2e$ $MJ^{-1}$). In contrast, Saudi Arabia and Norway have the lowest well-to-wake CI due to their domestic crude oil production being much less emissions-intensive than the global average and that 100% and 80% of their jet fuel consumption is met by domestic refining, respectively[34]. The US (90.8 $gCO_2e$ $MJ^{-1}$) and China (89.7 $gCO_2e$ $MJ^{-1}$), as shown in Fig. 1a, are the two biggest jet fuel producers and consumers, and their well-to-wake CIs are slightly higher than the global average (88.7 $gCO2e$ $MJ^{-1}$). The EU's average obtained from this study is 89.1 $gCO2e$ $MJ^{-1}$, which is comparable to the EU fossil fuel comparator (94.0 $gCO_2e$ $MJ^{-1}$, based on the marginal method, see Methods for more details)[39].

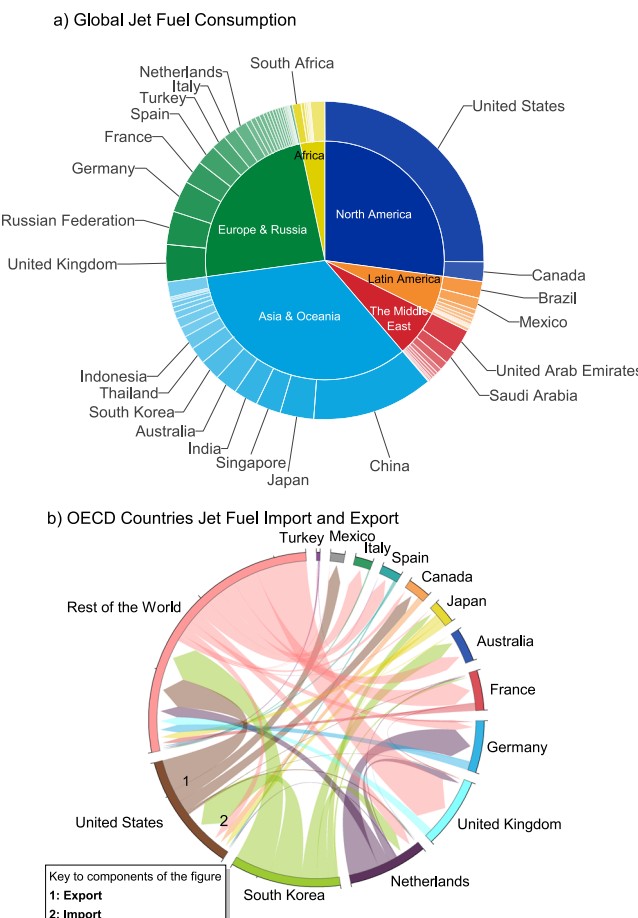

**Fig. 1 | Global jet fuel market in 2017. a** Global jet fuel consumption in 2017 (in barrels per day) by country and region. The inner pies are annual jet fuel consumption in percentage by world region. The outer arcs are annual jet fuel consumption in each country. The greater the arc length, the higher the consumption. The Middle East includes Iran, Saudi Arabia, UAE, Iraq, Qatar, Kuwait, Oman, Bahrain, Syria, Jordan, Lebanon, and Yemen. North America, Asia & Oceania, and Europe & Russia account for over 85% of global demand, with the US (25%), China (12%), and the UK (3.8%) topping the list. The remaining demand is spread across developed countries globally. The US (25%), China (13%), and South Korea (7%) are the top three producers. **b** Key Organisation for Economic Co-operation and Development (OECD) countries' annual jet fuel import and export. Countries are represented by outer segments arranged along the circle. Segment length is determined by the sum of export and import (from left to right of each segment, see the US example), which are depicted by arcs drawn between segments. The size of the arc is proportional to the weight of the flow. The arrowhead shows the direction of export. Key countries included in the rest of the world are China, India, Singapore, and Russia. Their exports are 7.8%, 8.2%, 8.4%, and 2.6% of the global export, whereas their imports are 4.4%, 0.3%, 3.7%, and 0.1% of the global import, respectively[62].

As shown in Fig. 2a, the volume-weighted average estimates made in this study are close to literature values reported by MIT[17], Zhou et al[18]., and the ICAO baseline[4]. However, the variation in the literature does not capture the full range of variation in emissions occurring in the jet fuel supply chain (see Supplementary Figs. 3 and 4). For example, at the country-level, crude oil production and refining emissions intensities range from 4.5 to 11.3 $gCO_2e$ $MJ^{-1}$ jet fuel and 1.6 to 11 $gCO_2e$ $MJ^{-1}$ jet fuel in the literature[18,20,21,24], respectively. At the same time, our study shows that they can vary from 2.9 to 27.6 $gCO_2e$ $MJ^{-1}$ jet fuel and 0.9 to 12.5 $gCO_2e$ $MJ^{-1}$ jet fuel, respectively (see Supplementary Table 1). Also, the well-to-wake CI calculated for jet fuel produced by each refinery has a much more noticeable variation (77.3-112 $gCO_2e$ $MJ^{-1}$) than the aggregated country-level results (81.1-94.8

$gCO_2e$ $MJ^{-1}$, Supplementary Fig. 2). This facility-level variation is mainly due to the production emissions associated with their crude slates and, more importantly, their choice of the primary stream of their jet fuel. Four main streams that can be blended into jet fuel in increasing refining CI are jet separated from the atmospheric crude distillation unit (CDU jet), jet from the kerosene hydrotreater (KHT jet), jet from the diesel hydrotreater (DHT jet), and jet from the distillate hydrocracking unit (DHCU jet). The stream-level and facility-level variations are almost identical, implying that the choice of jet fuel source can be a determining factor. For example, a refinery may produce jet fuel via more emissions-intensive processes (e.g., DHCU) to meet market demand or product quality standards (e.g., sulfur content). The choice of jet fuel source can also help SAF substitution that offers substantial climate benefits, depending on the fuel type (e.g., hydro-processed esters and fatty acids). The differences among these four refining pathways are discussed in more detail in the next section.

Figure 2b shows the country-level results by life cycle stage (see Supplementary Fig. 1). It estimates that the global volume-weighted average can be reduced by 2.1 and 7.1 $gCO_2e$ $MJ^{-1}$ to 86.6 and 81.6 $gCO_2e$ $MJ^{-1}$ under the low and high technology improvement scenarios, respectively (see Methods). This implies that a portion of current jet fuel production has the potential to meet the ICAO's sustainability criteria as LCAF with investment in supply chain decarbonization. However, it is worth noting that such potential is based on current crude oil refining throughput and may be subject to change of refinery operation modes (e.g., throughput change at refinery- and process unit-level and change of product slate to meet market demands) in the future.

**Well-to-Tank CI of Jet Fuel**

The well-to-tank portion of the life cycle includes well-to-refinery entrance, refining, and distribution. The global volume-weighted average well-to-refinery entrance CI is 9.6 $gCO_2e$ $MJ^{-1}$ jet fuel, comprised of 8.6 $gCO_2e$ $MJ^{-1}$ from crude oil production and 1.0 $gCO_2e$ $MJ^{-1}$ from crude oil transportation[40]. Country-level variations are 3.3-27.6, 2.9-27.6, and <0.1-2 $gCO_2e$ $MJ^{-1}$, respectively (Supplementary Table 1). Such large variations (mainly stemming from crude oil production) indicate a high correlation between the well-to-wake and well-to-refinery entrance CI ($R = 0.83$). Note that the global volume-weighted average well-to-refinery entrance CI can be reduced by 1.6 and 4.5 $gCO_2e$ $MJ^{-1}$ under the low and high technology improvement scenarios, respectively. More discussion on crude oil production emissions has been investigated in previous work[41].

The global volume-weighted average refining CI is 4.5 $gCO_2e$ $MJ^{-1}$ jet fuel and agrees well with literature values (see Supplementary Figs. 5 and 6). Although moderately correlated with the well-to-wake CI ($R = 0.34$), a sensitivity analysis shows that the refining CI is affected by the allocation method applied, the energy use of key process units, and the indirect emissions of natural gas (see Supplementary Fig. 7). It is worth noting that switching from the process unit-level (adopted as the default in this study) to the refinery-level allocation can drastically increase jet fuel refining CI because jet fuel, in general, is processed by less units than other major refinery products (e.g., gasoline and diesel). Fig 3a presents refining CI by jet fuel source (i.e., characterized by crude oil fractions and process units). Deep conversion refineries that process more heavy crude oils are in general more energy- and emissions-intensive than medium conversion and hydroskimming refineries that generally handle lighter crude oils[34]. Each jet fuel source also has different refining CI. DHCU jet has the highest refining CI, ranging from 9 to 11.5 $gCO_2e$ $MJ^{-1}$ depending on refinery configuration, followed by DHT (3.4-7.5 $gCO_2e$ $MJ^{-1}$) and KHT jet (3.1-4.8 $gCO_2e$ $MJ^{-1}$), whereas CDU jet has the lowest CI, approximately 1 $gCO_2e$ $MJ^{-1}$. DHCU jet, DHT jet, and KHT jet are processed by more units and thus require more hydrogen and heat than CDU jet. As a result, the amounts of jet fuel produced from DHCU and CDU are

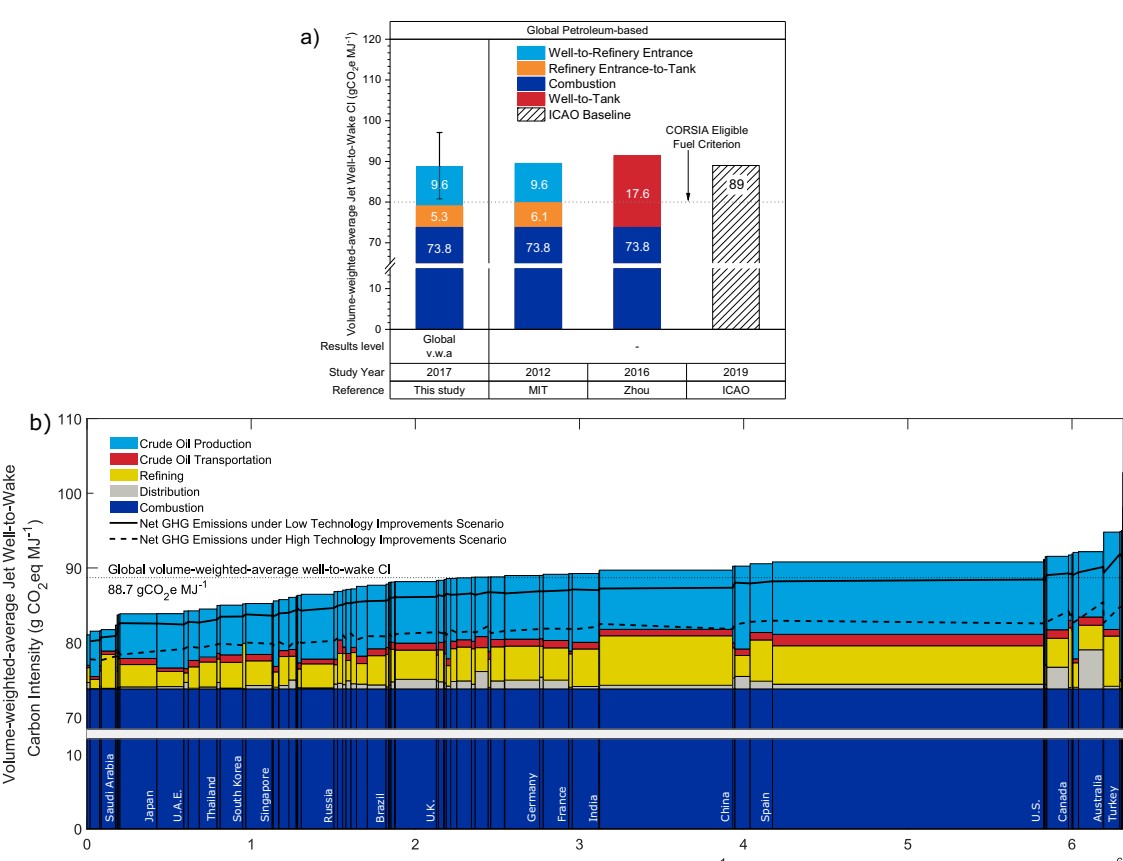

**Fig. 2 | Well-to-wake carbon intensity of jet fuel. a** Comparison between results obtained from this study and jet fuel well-to-wake CI values reported in the literature. Well-to-refinery entrance includes crude oil production and transportation. Refinery entrance-to-tank includes refining and distribution. Study year refers to the time period when data were collected (not the year the study was published). Global v.w.a. represents the global volume-weighted average CI (88.7 gCO$_2$e MJ$^{-1}$). Whiskers are 5 and 95 percentiles of the well-to-wake CI of jet fuel produced by refineries. See references for studies conducted by MIT[17] and Zhou et al[18]. and note that average combustion emissions are adjusted to 73.8 gCO$_2$e MJ$^{-1}$ for literature studies presented here. Energy-based process unit-level based allocation is adopted for refining in this study (see Methods). **b** 2017 country-level volume-weighted average jet fuel well-to-wake CI supply curve and emissions reduction scenarios.

Each vertical stacked bar represents a life cycle stage breakdown of the volume-weighted average well-to-wake CI of one country (sorted by increasing CI value). The width of the stacked bar indicates the daily volume of jet fuel consumed in the corresponding country. The low technology improvements scenario assumes minimal routine flaring in crude oil production and transportation, and carbon capture from the fluid catalytic cracker and steam methane reformer (SMR for hydrogen production) units. The high technology improvements scenario assumes no routine flaring and minimal fugitives & venting in crude oil production and transportation, and carbon capture from all refinery process units and the use of low-carbon steam and electricity for refinery operations[34,41]. In addition, all carbon capture processes are assumed to be powered by low-carbon utilities. See Methods for details about these two scenarios.

most influential to a refinery's jet fuel CI (see Supplementary Fig. 8 and Supplementary Table 2). SMR hydrogen production is one of the most energy- and emissions-intensive refinery processes. It represents almost 60% of the refining CI of jet fuel produced from DHCU but less so for those from DHT and KHT because hydrocracking (e.g., DHCU) in general consumes more hydrogen than hydrotreating (DHT and KHT). The global volume-weighted average refining CI is estimated to have the potential to be reduced to 4 and 2 gCO$_2$e MJ$^{-1}$ under the low and high technology improvement scenarios, respectively (Supplementary Fig. 9).

As shown in Fig. 3b, CDU jet and KHT jet are the least emissions-intensive streams and are responsible for more than 60% of the global jet fuel demand and approximately 70-80% of the jet fuel demand in Asia and Oceania, Latin America, and Africa. On the other hand, DHT jet and DHCU jet play a key role in Russia, Europe, and the Middle East (see Supplementary Tables 3 and 4). Over 90% of the global jet fuel production by volume is from medium and deep conversion refineries. Hydroskimming refineries generally produce jet fuel from the CDU unit. In contrast, deep conversion refineries produce more KHT jet and DHCU jet. One may argue that all refiners should produce as much CDU jet as possible to reduce the carbon footprint of jet fuel. However,

refineries, in general have limited control over the composition of jet fuel, which is primarily determined by the quality of crude supply and required specification of all products via product blending. Supplementary Table 5 confirms that the greater the refinery complexity, the less likely refineries are solely dependent on CDU jet. The portion of refineries that only produce KHT and DHT jet, in contrast, seems irrelevant to refinery complexity. From a regional perspective, 39-56% of refineries in Asia and Oceania, Latin America, and Africa only supply CDU jet to the market, higher than the global average of 29%. This regional difference may be explained by differential fuel quality requirements such that the severity of hydrotreating may vary.

The jet fuel distribution CI can vary from <0.1 to 5.2 gCO$_2$e MJ$^{-1}$ at the country-level with a global volume-weighted average of 0.8 gCO$_2$e MJ$^{-1}$. The distribution CI is greater than 2 gCO$_2$e MJ$^{-1}$ in Australia, Canada, and Mexico (mainly due to long distances between refineries and airports) and less than 1 gCO$_2$e MJ$^{-1}$ in ~80% of the countries in this study. This value can be extremely low depending on the proximity of a refinery or a port to an airport (e.g., a few kilometers of pipeline transport from refinery to airport). The reduction potentials under the low and high technology improvement scenarios are ≤0.1 gCO$_2$e MJ$^{-1}$.

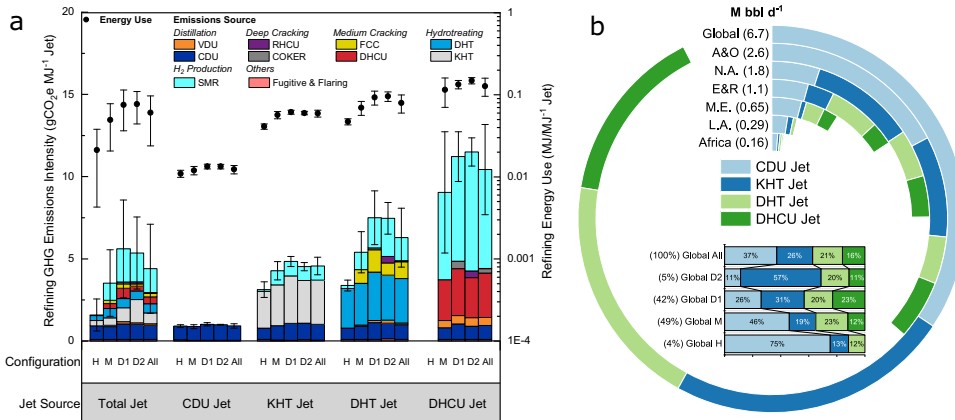

**Fig. 3 | Jet fuel refining CI breakdown and jet fuel composition. a** Volume-weighted average jet fuel refining energy use and CI breakdown by process unit, refinery configuration, and jet fuel source. Each stacked bar (breakdown by process unit) represents the volume-weighted average CI of a given source of jet fuel produced by a specific type of refinery configuration. Whiskers are volume-weighted average standard deviation. Energy use (secondary y-axis on a log scale) includes electricity, steam, heat, hydrogen, and FCC coke burn. Emissions from each process unit are calculated by converting the energy use (i.e., the portion allocated to jet fuel) to GHG emissions using emissions factors adopted in PRELIM. SMR: steam methane reformer; RHCU: residual hydrocracking unit; FCC: fluid catalytic cracking unit; DHT: diesel hydrotreater; DHCU distillate hydrocracking unit; KHT: kerosene hydrotreater; VDU and CDU: vacuum and atmospheric distillation units, respectively. Configurations H, M, D1, D2, and All

are hydroskimming, medium conversion, deep conversion coking, deep conversion hydrocracking, and all refineries, respectively (see Supplementary Note 1 for configuration definitions). Jet source represents the crude fractions and refinery streams blended into jet fuel exiting the refinery. **b** Jet fuel source volumetric composition by region. A&O is Asia and Oceania. NA and LA are North America and Latin America, respectively. E&R is in Europe and Russia. ME is the Middle East. Arc length shows the contribution of each source to the jet fuel produced in each region. The inset stacked bar chart shows the global average composition of jet fuel produced by hydroskimming (H), medium conversion (M), deep conversion coking (D1), deep conversion hydrocracking (D2), and all refineries (All). Percentages in parentheses next to configuration labels are the volume shares by refinery type (e.g., hydroskimming refineries only produce 4 vol.% of global jet fuel).

## Combustion

Although this study adopts the ICAO baseline of 73.8 gCO₂e MJ⁻¹ for jet fuel combustion, a sensitivity analysis on refinery-level results (PRELIM tracks the carbon content of jet fuel produced) shows that jet fuel combustion emissions can vary from 72.8 to 75.4 gCO₂e MJ⁻¹ with a weighted average of 74.0 gCO₂e MJ⁻¹. Future work should investigate this variation by sampling refinery products for combustion testing.

## Policy implications

As shown in Fig. 4, the cumulative well-to-wake baseline emissions for international aviation from 2019 to 2050 are ~44 GtCO₂e (i.e., the area between the baseline curve and the horizontal axis). These emissions are equivalent to between 8% and 10% of the remaining carbon budget if a 50% and 66% chance of limiting global warming to 1.5 °C is considered, respectively[5]. To meet the ICAO's carbon-neutral growth goal, an aggressive reduction of 21.6 GtCO₂e cumulative emissions must be achieved by 2050. This presents a critical challenge to the aviation industry and requires rapid deployment of a variety of strategies, such as technology development, operational improvements, carbon offsetting, and low-carbon aviation fuels that meet specific sustainability criteria (see Supplementary Note 2 for details).

Technology and infrastructure improvements are expected to reduce 7.8 and 2.7 GtCO₂e emissions, respectively. Ideally, if the remaining reductions (11.1 GtCO₂e) are achieved by SAF, with core LCA values of 10.4 and 40.4 gCO₂e MJ⁻¹ (Supplementary Table 9, Herbaceous energy crops and Soybean oil used as two examples of CORSIA eligible fuels), then 29% and 47% of the cumulative jet fuel demand (3.3 and 5.4 Gt jet fuel, respectively) by 2050 needs to be displaced, respectively. Despite its rapid development, SAF is not forecast to scale at a rate and magnitude to fully replace petroleum jet fuel in the near term[42–44]. In addition, biomass-based SAF have raised questions about feedstock availability, land availability, conversion efficiency, and competition with other sectors (e.g., road transport and chemical market)[45]. The spatial variation of jet fuel consumption and supply chain emissions among different regions further complicates this

challenge and calls for a clearer understanding of how SAF can be best deployed in different regions to maximize emissions reductions. It has been recently reported that EU is likely to shift more biofuels from road transport to aviation[46–48]. Future work needs to assess other emerging technologies such as power-to-liquid SAF and hydrogen fuel cells[49,50].

This study fills an important knowledge gap by presenting the first life cycle assessment of global jet fuel well-to-wake GHG emissions using a bottom-up engineering-based modeling approach. Engineering-based tools, including OPGEE, COPTEM, and PRELIM are employed to quantify emissions in the well-to-refinery exit stage. Emissions from the distribution and combustion stages are calculated based on the average minimum refinery-to-airport distance and literature recommendations. The global volume-weighted average well-to-wake CI is 88.7 gCO₂e MJ⁻¹ jet fuel. This average value agrees with the ICAO baseline and other literature. We also, for the first time, provide detailed country-level CI variation that ranges from 81.1 to 94.8 gCO₂e MJ⁻¹ jet fuel, which is mainly attributed to the variation in crude oil production emissions. With proper supply-chain decarbonization, up to 2.3 GtCO₂e emissions can be avoided by 2050. Having such a transparent jet fuel CI baseline can improve decision-making amid the transition to decarbonizing aviation by understanding the key emissions drivers and designing policy incentives (e.g., prioritizing the replacement of the most emissions-intensive petroleum jet fuel pathways with SAF). It should be noted that the bottom-up attributional LCA approach adopted in this study provides an average baseline of jet fuel supply chain footprint. However, such an approach is not intended to be used to evaluate the indirect effects of changing markets and policies, which should be considered in future research. Other limitations and future work include: (1) regularly updating the latest crude oil production and transportation carbon intensity, (2) differentiating each individual refinery in PRELIM by matching the actual throughput and energy use of key process units, and (3) comparing the costs of decarbonization options of the petroleum supply chain with those of SAF to make economically sound decisions.

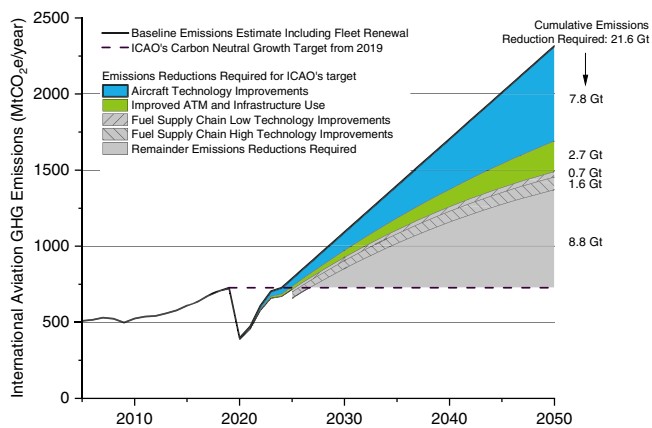

**Fig. 4 | Well-to-wake GHG emissions and reductions potential from international aviation 2019-2050.** Only international aviation is considered. Jet fuel consumption data are from IEA (2005-2024)[32] and International Civil Aviation Organization (ICAO, 2050)[4]. The original IEA data includes domestic and international consumption; therefore, a 57% share is calculated for the international portion based on the 2017 record and assumed for the whole period. The base case assumes a constant well-to-wake CI of 3.8 tonne $CO_2e$ per tonne jet fuel (equivalent to 88.7 $gCO_2e$ $MJ^{-1}$ jet fuel presented above for 2017) and a cumulative consumption of 11.5 Gt jet fuel over the period from 2019 to 2050, indicating that the cumulative emissions are 44 $GtCO_2e$ (2019-2050). Also, this base case estimate considers the impact of fleet renewal with increasing fleet efficiency and therefore reduced jet fuel consumption. Aircraft Technology improvements aim at propulsion, aerodynamics, weight reduction, and urban air mobility. ATM stands for air traffic management. Infrastructure use refers to operational improvement. These two (starting from 2019, regardless of jet fuel demand drop due to COVID-19) can improve fuel efficiency and reduce fuel consumption by 1.37% per annum (0.98% from technology and 0.39 from operations)[4]. Therefore, their potential impacts on emissions reductions from 2019 to 2050 are 7.8 and 2.7 $GtCO_2e$, respectively. The low and high technology improvement scenarios target carbon capture and low-carbon utilities, which are the same as explained in Fig. 2c and Methods. They can reduce 0.7 and 2.3 $GtCO_2e$ by 2050 (starting from 2026 when the predicted annual emissions less technology and infrastructure improvements are above the ICAO's carbon-neutral growth target), respectively (the low technology improvements scenario is a subset of the high technology improvements scenario). The remaining emissions reductions required from SAF and carbon offsets are 8.8 $GtCO_2e$ (11.1 $GtCO_2e$ if no supply chain technology improvements are adopted). The carbon-neutral growth target line (based on the 2019 pre-pandemic level) requires 21.6 $GtCO_2e$ emissions reductions by 2050. Note the original emissions baseline for ICAO's carbon-neutral growth target is the average of 2019 and 2020, which was amended to the 2019 emission levels due to COVID-19[63]. See more details in Supplementary Note 2. Other aspects not included in this forecasting: 1) advances in crude oil extraction and refinery operations that are more energy efficient and 2) shift in refinery product demand.

## Methods

### Data sources and data processing

Global jet fuel production, export, demand, and import in 2017 (Fig. 1) are obtained from IEA World Oil Statistics[32] and GlobalData[33]. For jet fuel life cycle CI estimates, detailed information about crude production, transportation, refining, and jet fuel distribution is needed. Not all jet fuel demand reported by IEA can be tracked. In this study, Wood Mackenzie commercial datasets[30,31] are purchased for estimating jet fuel life cycle CI. These datasets represent 94% (6.3 Mbbl $d^{-1}$) of the global jet fuel demand in 2017 (6.7 Mbbl $d^{-1}$) reported by IEA[32] and GlobalData[33]. Therefore, the results are believed to be estimated from a representative global data source.

To calculate jet fuel life cycle GHG emissions, we adopt a process-based attributional LCA approach that accounts for mass and energy flows along its supply chain. It is comparable to the CORSIA methodology for calculating life cycle emissions values of CORSIA eligible fuels[37,51]. Supplementary Fig. 1 presents the LCA system boundary of jet

fuel, including crude oil production, crude oil transportation, crude oil refining, jet fuel distribution, and jet fuel combustion. Life cycle GHG emissions are calculated by employing various engineering-based life cycle assessment tools (i.e., crude oil production, transportation, and refining) and applying emissions factors to fuels consumed (i.e., jet fuel distribution and combustion). Note that upstream emissions associated with materials and energy consumption (e.g., electricity production and natural gas extraction & processing) are accounted for in all stages.

Crude oil production (i.e., extraction and processing) emissions are adopted from Masnadi et al[41]., in which the Oil Production Greenhouse gas Emissions Estimator (OPGEE v2.0) was applied. Emissions incurred during crude oil transportation, including pipeline transportation from field to terminal, marine shipping/pipeline transportation between terminals, and pipeline transportation from the terminal to refinery entrance, are adopted from Dixit et al[40]. in which crude oil pipeline transportation emissions model (COPTEM) was used for calculating pipeline transportation GHG emissions[28]. Crude oil production and transportation emissions are in $gCO_2e$ $MJ^{-1}$ crude and can be converted to $gCO_2e$ $MJ^{-1}$ jet fuel by multiplying by the crude refining efficacy in MJ crude $MJ^{-1}$ jet fuel.

Emissions from refining crude oil into jet fuel are estimated using PRELIM v1.5. Data associated with 354 marketable crude oils (that is, source country, assay profile including API gravity and sulfur/nitrogen contents) and 481 refineries (that is, location, crude input, process unit capacities, Nelson complexity, intermediate flow split ratios, and flow rates, intermediate flow properties, product blending, and utility consumption) are obtained from commercial datasets[30,31]. These commercial datasets are not available in the public domain. Crude assays collected from the commercial datasets are transformed into a PRELIM-compatible format that comprises nine temperature fractions (i.e., cuts) from light straight run to vacuum residue. This transformation procedure includes prediction of crude distillation curve fitting, PRELIM fractions determination, and fractional specification (e.g., API gravity, sulfur content, nitrogen content, hydrogen content, characterization factor). Details of the transformation procedure can be found in the PRELIM documentation, and a MATLAB-based app is available for download from the PRELIM website[29]. Jet fuel refining CI is in $gCO_2e$ $MJ^{-1}$ jet fuel.

The average minimum distance between refineries and airports in each country is calculated based on their coordinates for domestic jet fuel distribution from refinery to aircraft fuel tank. The coordinates of refineries are from Wood Mackenzie commercial datasets[30,31], and the coordinates of airports are publicly available[52] (lists of selected refineries and airports are available upon reasonable request). If we assume 60 $gCO_2e$/tonne-km via heavy-duty vehicle[53] and 46000 MJ/tonne jet fuel, then a ground transportation CI of 0.0013 $gCO_2e$/MJ jet fuel-km can be derived. Jet fuel distribution CI is in $gCO_2e$ $MJ^{-1}$ jet fuel.

Jet fuel combustion emissions factor is set as 73.8 $gCO_2e$ $MJ^{-1}$ jet fuel according to International Civil Aviation Organization (ICAO, 3.16 kg $CO_2$ from 1 kg jet fuel and 42.8 MJ $kg^{-1}$ jet fuel)[4,54].

Jet fuel life cycle emissions (well-to-wake) are summarized and interpreted at both refinery and country levels. The refinery-level results include the life cycle emissions of jet fuel produced by each refinery and consumed within the country in which this refinery is located. For the country-level results, domestic refinery-level well-to-wake results and emissions associated with imported jet fuel are accounted for (22 countries with no import). For jet fuel import and export, the global average well-to-refinery entrance and refining emissions of jet fuel are assigned. The average minimum distance between refineries and terminals in jet fuel-producing countries and the average minimum distance between terminals and airports in fuel-consuming countries are calculated based on their coordinates. The coordinates of terminals are publicly available[55], and a list of selected coordinates for this study is available upon reasonable request. The

transportation of jet fuel between terminals, either via marine shipping or pipeline, is calculated using the global average estimated above.

Note for crude oil production and transportation, greenhouse gases included are $CO_2$, $CO$, $CH_4$, $N_2O$, and non-methane volatile organic compounds (NMVOCs); for crude oil refining, greenhouse gases included are $CO_2$, $CH_4$, $N_2O$, and non-methane volatile organic compounds (NMVOCs); for jet fuel combustion, according to ICAO's jet fuel conversion factor, only $CO_2$ is included.

## OPGEE

The Oil Production Greenhouse Gas Emissions Estimator (OPGEE v2.0) is an engineering-based life cycle assessment (LCA) tool for evaluating GHG emissions from crude oil production, processing, and transport. OPGEE is built in Excel using engineering fundamentals, ensuring transparency, flexibility, and accuracy. Its system boundary starts from initial exploration and ends at the refinery entrance with a functional unit of 1 MJ of petroleum crude. OPGEE considers emissions from on-site fuel combustion, indirect emissions associated with electricity import and fuel production and transport, flaring and fugitive emissions, and embodied emissions from on-site materials. More details about OPGEE and its settings can be found in the literature[41].

## COPTEM

The fluid mechanics-based Crude Oil Pipeline Transportation Emissions Model (COPTEM) is used to estimate the GHG emissions associated with crude oil transport via pipeline[28]. It is designed to accommodate a broad range of oil properties and pipeline dimensions. The system boundary of COPTEM is drawn in between two given crude oil terminals connected by pipeline. Its functional unit is $kgCO_2e \ bbl^{-1}$ for a given length of the pipeline. COPTEM is built upon the Energy, Darcy-Weisbach, and Colebrook-White equations, which collectively can be seen as a function of crude oil parameters, pipeline dimensions, and external factors. It also considers the impact of heat transfer between transported crude and the ambient environment on oil viscosity. The energy requirement and GHG emissions associated with maintaining a sufficient hydraulic head can be calculated accordingly. More details about this model and the settings for this study can be found in the literature[28].

## PRELIM

PRELIM v1.5 is used in this study. PRELIM is an Excel-based open-source tool that examines how refinery energy use and GHG emissions can vary based on different crudes and refinery configurations. It was developed by the Energy Technology Assessment Research Group at the University of Calgary and has since been continuously updated[29]. PRELIM is bottom-up in construction and tracks the flows in and out of each process unit and energy use by each process unit via a refinery linear programming modeling method. There are 11 configurations available in PRELIM, which can be further grouped into four main types: hydroskimming, medium conversion, deep conversion coking, and deep conversion hydrocracking (see Supplementary Note 1). Four different allocation methods (i.e., mass-, energy-, market value-, hydrogen content-based) are available for allocating energy use and GHG emissions (converted from energy use by emissions factors) to various intermediate streams at the process-unit level and eventually to final products.

The system boundary in PRELIM is refinery-entrance to refinery-gate, including all process units in a refinery and energy consumed (i.e., electricity, heat, steam, and hydrogen). Aside from direct emissions (e.g., onsite hydrogen generation), indirect emissions from the extraction, processing and transportation of energy products (i.e., natural gas and electricity) are also considered. The functional unit can be either 1 bbl crude or 1 MJ final product. Key settings and updates in PRELIM v1.5 are listed below (more details can be found in SI):

1. As compared to earlier versions of PRELIM in which jet fuel is solely produced via kerosene hydrotreater (hydrotreating the kerosene cut from the atmospheric tower – CDU jet), PRELIM 1.5 now has three more sources of jet fuel, including direct separation of CDU jet without hydrotreating, production from diesel hydrotreater (DHT jet), and production from distillate hydro-cracking unit (DHCU jet). Key process unit yields, flow split ratios, and jet fuel blending of all 481 refineries are closely matched with the commercial datasets[30,31]. Given different sources of jet fuel carry different amounts of emissions burden, matching the contributions of each source to jet fuel blending is the key to understanding the variation of jet fuel refining CI. In addition, the refinery configurations available in PRELIM are typical and are not intended to represent any specific refinery. Therefore, in this study, commercial datasets calibrate each refinery in PRELIM to match product slate, process unit capacity constraints, and jet fuel composition. PRELIM's product slate considers 16 products, including gasoline, jet fuel, diesel, fuel oil/furnace oil, coke/residue, liquid heavy ends/bunker C, sulfur, refinery fuel gas, surplus reformer hydrogen, liquified petroleum gas, petrochemical feedstocks, asphalt, atmospheric residue, naphtha, light vacuum gas oil, and heavy vacuum gas oil.
2. The indirect emissions associated with natural gas are country-specific for countries listed in Supplementary Table 6. Countries not included in this list have the default indirect emissions of 10.8 $gCO_2e \ MJ^{-1}$ natural gas.
3. Electricity grid emissions intensity of each country is adopted from IEA[56]. An example is shown in Supplementary Table 7.
4. Cogeneration of heat and power (CHP) is considered for all 481 refineries based on publicly available information. In total, 15% of the refineries have natural gas combustion turbines & heat recovery steam generators, 20% of the refineries have a natural gas combined cycle, 20% of the refineries have boiler & steam turbines, and the remaining 45% of refineries that have no such information available are deemed to be no-CHP. The detailed list is available from the corresponding author upon reasonable request.
5. According to commercial datasets[31], refineries have four FCC hydrotreating options (i.e., FCC feed hydrotreater, FCC naphtha/gasoline hydrotreater, both, and none).
6. The process-unit-level energy-based allocation method is chosen because it aligns well with recent studies on the life cycle emissions of energy systems in which the energy-based allocation was applied[34,41,57]. Also note that the most recent CORSIA methods for calculating actual life cycle emissions values also adopted process-unit-level energy-based allocation method[37]. All products (i.e., 16 in total) are included for allocation and a variety of allocation methods are tested in the sensitivity analysis.
7. Process unit energy (i.e., electricity, heat, steam, and hydrogen) consumption rates are updated using average regional values[31]. There are six regions included: North America, Latin America, The Middle East, Asia & Oceania, Europe & Russia, and Africa.
8. Lower heating value (LHV) is employed. A third-order polynomial equation from American Petroleum Institute is adopted as the default LHV calculation method.
9. Naphtha catalytic reformer only processes straight-run naphtha.
10. Pressure swing adsorption (PSA) is adopted as the hydrogen purification method for steam methane reformer (SMR).
11. The AR5 2013 IPCC 100-year GWP emissions factors are employed.
12. Refinery fuel gas produced onsite is combusted as fuel or used as SMR feedstock.
13. Hydrogen demand is assumed to be satisfied by the onsite catalytic naphtha reformer and SMR.

Crude feedstock is blended into an unique crude blend for each refinery and fed into PRELIM. All model settings are adjusted as stated

above. Results obtained from the PRELIM run of each refinery are: jet fuel production rate (MJ d$^{-1}$ and bbl d$^{-1}$), jet fuel refining crude requirement (MJ crude MJ$^{-1}$ jet fuel), jet fuel refining energy consumption (MJ energy MJ$^{-1}$ jet fuel), jet fuel refining CI (gCO$_2$e MJ$^{-1}$ jet fuel), and jet fuel composition (CDU jet, KHT jet, DHT jet, and DHCU jet in vol.%).

We continue to update PRELIM to improve the accuracy of its estimates as better data (e.g., process unit energy consumption, intermediate material flows, and product blending) become available. Nonetheless, there are some inherent limitations associated with PRELIM: 1) jet fuel produced from straight run naphtha and FCC Naphtha are not considered; 2) hydrogen is assumed to be produced onsite with production emissions from a steam methane reformer rather than purchased. These will be addressed in future versions of PRELIM.

### Jet fuel supply chain GHG emissions reduction technologies and scenarios

Emissions reductions from the production and transportation of crude oil include energy efficiency improvement, carbon capture (CC), flaring management, leak detection and repair, etc. Emissions reduction technologies adopted for refining include post-combustion CC and low-carbon utilities (e.g., electricity and steam from solar energy). It is assumed that all CC equipment is powered by low-carbon utilities.

In this study, for the refining stage under the low technology improvements scenario, we assume that post-combustion CC is only applied to fluid catalytic cracking unit (FCC) and steam methane reformer (SMR) for all refineries. This is because these two are usually the most emissions intensive units across any given refinery and it is technically feasible to install CC equipment under most circumstances. FCC in general converts heavy oil fractions to value-adding light products that can be either blended as gasoline and diesel or processed for producing LPG and petrochemical feedstock. The most important emissions source of FCC is the carbonaceous deposit (coke) on the surface of the catalyst that needs to be burnt off regularly. On the other hand, SMR produces hydrogen from natural gas and high temperature steam (also produces CO$_2$ as a by-product). It is the main hydrogen source that provides hydrogen for hydroprocessing and is usually responsible for 30%-50% of the GHG emissions from a refinery. The reaction process is endothermic, which means there needs to be a heat source. In PRELIM, we assume natural gas and refinery fuel gas are combusted to satisfy this heat requirement. Therefore, CO$_2$ from both combustion and chemical conversion of natural gas and refinery fuel gas are available for CC.

For the refining stage under the high technology improvements scenario, in addition to CC being applied to FCC and SMR, we assume CC is also applied to capture and sequester CO$_2$ from natural gas combustion for providing heat to key process units including atmospheric tower, vacuum tower, coker, residue hydrocracker, gasoil hydrocracker, all hydrotreaters, naphtha reformer, desalter, kerosene merox unit, alkylation unit, isomerization unit, and fuel gas treatment unit. In addition, low-carbon electricity and steam are adopted for crude oil refining. Low-carbon electricity is available from resources such as nuclear- or renewable-powered electricity generating facilities, while lower-carbon steam may be generated from CHP systems or renewable driven heat-generating facilities. Emissions not captured (but counted as refining emissions) are indirect (Scope 2) emissions associated with electricity and natural gas production, and support services emissions (flaring and fugitive emissions).

We assume an amine-based post-combustion CC with an 85% capture efficiency as suggested by literature recommendations[58–60]. Hot flue gas captured from combustion processes passes through a waste-heat steam generator first, followed by CO$_2$ absorption in aqueous monoethanolamine. The absorption medium can be regenerated via heat. To meet water and pressure specifications, CO$_2$ compression and dehydration are also required. Energy (5.97 MJ steam and 0.2 kWh electricity per kg capture CO$_2$ as suggested by Motazedi et al., 2017[61]) required by the CC equipment is assumed to be provided by low-carbon electricity and steam as described above.

For emissions reductions associated with crude oil production and transportation, based on literature recommendations[41], we assume a 16.7% emissions reduction through routine gas flaring minimization for the low technology improvement scenario. We assume a 47% emissions reduction through no routine gas flaring and minimal fugitives & venting for the high technology improvement scenario. See previous studies for more details[34,41].

### Sensitivity analysis

A sensitivity analysis is conducted to study the importance of PRELIM parameters on carbon intensity estimates. The following parameters are selected based on our previous studies: natural gas indirect (e.g., extraction and processing) emissions factor, process energy required by key process units, SMR hydrogen production emissions, and the allocation method. Baseline values of these parameters listed in Supplementary Table 8 are PRELIM defaults used for the base case presented in the main text. Parameter variations are set by literature recommendations to ensure as much geographical representation as possible.

We choose 100 random refineries for this sensitivity analysis by considering time and resource constraints. Their baseline volume-weighted average jet fuel refining CI is 4.02 gCO$_2$e MJ$^{-1}$. For each run, we compare the volume-weighted average jet fuel refining CI to the base case.

Note that the choice of allocation (or displacement or marginal) method for co-product treatment can be arbitrary and therefore may have a substantial impact on the refining CI of jet fuel, leading to different results and interpretations. It can be misleading if mitigation options are compared to a baseline that is not consistent in its methods as negative unintended consequences can occur. As such, GHG reduction policies and strategies must be transparent about the methods and data used to calculate the baseline and ensure that any mitigation opportunity is based on application of consistent methods.

### Statistical tests used

All data processing and data analyses are carried out using MATLAB R2019a. Pearson correlation coefficient is used to measure the degree of correlation with $P < 0.05$ indicating statistical significance.

## Data availability

The data that support the findings of this study are available from Wood Mackenzie (i.e., crude oil production, transportation, and refining), International Energy Agency (i.e., jet fuel market analysis), public statistics, scientific/technical papers (i.e., crude oil production), and publicly available information (i.e., refinery, airport, and terminal coordinates for jet fuel distribution). The Wood Mackenzie and IEA data are available under restricted access for the current study and therefore they are not publicly available. Access may be obtained from Wood Mackenzie and IEA.

## Code availability

We use OPGEE 2.0, COPTEM v1.0, and PRELIM v1.5 for this study. OPGEE 2.0 is publicly available from https://eao.stanford.edu/research-areas/opgee. COPTEM v1.0 is available from the corresponding author upon reasonable request. PRELIM v1.5 is publicly available from https://ucalgary.ca/energy-technology-assessment/open-source-models/prelim. The MATLAB codes for aggregating and processing OPGEE, COPTEM, and PRELIM outputs are available from the corresponding author upon reasonable request.

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

## Acknowledgements
Aramco Americas (MI, USA) provided financial support to L.J. Special thanks go to the Natural Sciences and Engineering Research Council of Canada (NSERC).

## Author contributions
L.J., H.M.E.-H., J-C.M., and J.A.B. were involved in data gathering, data processing, and modeling framework design. L.J. integrated the final modeling results and wrote the manuscript. L.J., H.M.E.-H., J-C.M., J.L., A.S.Q., Y.D., R.S., A.R.B., M.S.M., H.L.M., W.P., D.G., and J.A.B. contributed to revising the paper.

## Competing interests
Prof. Bergerson is funded by Aramco Americas. The remaining authors declare no competing interests.
