## [Peer Review File · Nature Communications]

Understanding variability in petroleum jet fuel life cycle
greenhouse gas emissions to inform aviation decarbonizationREVIEWER COMMENTS

Reviewer #1 (Remarks to the Author):

This study provides a noteworthy estimation of life cycle GHG emissions from petroleum jet fuel at the regional level.

The article is well-written, original and scientifically valuable and has a higher standard than the average articles published in this field.

I think that the significance of the findings in this field is high but probably not or just enough to be published in Nature Communications (also in light of the similar article recently published in Nature Climate change "Carbon intensity of global crude oil refining and mitigation potential").

The work with the respective methodology supports well the conclusions and claims.

The detail provided in the methods is enough transparent to allow reproduction.

Reviewer #2 (Remarks to the Author):

Nature Communications manuscript NCOMMS-22-22014-T

Key results

The authors conducted a detailed and thorough analysis of the well-to-wake greenhouse gas (GHG) footprint of global jet fuel consumption based on a process-based attributional life cycle assessment approach. They calculated a global-weighted average of 88.7 g CO₂eq MJ⁻¹ with ranges of 81.1 to 94.8 g CO₂eq MJ⁻¹ at the country level. Whilst emissions of combustion are constant, the variation upstream in the supply chain from crude oil production, transportation and refining provides important insights in the GHG footprint of the aviation sector and options and limitations to reduce it. To this purpose, the study also includes two future scenarios that explore GHG savings from technological development.

Validity

A sensitivity check is included for key processes and results are correctly validated. However, there are a few points concerning the interpretation phase of LCA:

- Firstly, the limitations of the method and input data need some more elaboration. For example, the chosen allocation method of upstream emissions in the refinery (energy allocation) has a significant impact on the results; see Moretti et al. (2017).
 - Secondly, the goal and scope of the study, applied method and results for the current well-to-wake GHG emissions are correctly aligned. However, the study also draws some conclusions and makes ambitious promises in the title (climate-neutral aviation) about the future (2050) that are inconsistent with the inventory data that is not representative for refineries operating in a climate-neutral energy sector. They will need to adjust their outputs to changing market demands.
- Moretti, C., Moro, A., Edwards, R., Rocco, M.V. and Colombo, E., 2017. Analysis of standard and innovative methods for allocating upstream and refinery GHG emissions to oil products. *Applied Energy*, 206, pp.372-381.

Significance

The results are significant, but the title, conclusions, and recommendations must be consistent with the limitations of the method and input data.

Data and methodology

The authors provide a good description of the data sources and models used in the supplementary information and offer additional data upon request, which will be helpful to modelers and other practitioners who will use this manuscript. The method section has a few limitations that should be easy to address:

- The characterization of GHG emissions is not explained. If non-CO₂ GHG emissions are ignored, both the chosen scope and the implications to the results should be clearly explained in the main text.
- The explanation of the scenarios in the supplementary information is not very clear. A table with

the scenario assumptions supported with argumentation would be helpful.

Analytical approach

No comments.

Suggested improvements

L1-2: the title is too ambitious regarding climate-neutral aviation and content covered in the manuscript. Optimizing fossil jet supply will not be sufficient to meet CORSIA objectives (climate-neutral growth).

L306: the section on policy limitations should also cover the limitations of attributional modeling concerning (policy) decision support and change-oriented scenario analyses.

L213-214: a critical note on these investments, risks of lock-in in current refinery capacity, and the changing role of refineries in the energy transition would be helpful.

L316-327: it would be interesting to cover the challenges of the transition of the aviation sector in more detail. For example, liquid electrofuels (PtX) are not included in the text. Furthermore, significant amounts of biofuels will likely shift from road transport to aviation under updated support policies in the EU (RED II and RED III). See, for example, de Jong et al (2018). de Jong, S., van Stralen, J., Londo, M., Hoefnagels, R., Faaij, A. and Junginger, M., 2018. Renewable jet fuel supply scenarios in the European Union in 2021–2030 in the context of proposed biofuel policy and competing biomass demand. *Gcb Bioenergy*, 10(9), pp.661-682. Finally, a recommendation for further research beyond the scope of the presented manuscript would be to assess marginal emissions of current and future jet fuel supply.

Clarity and context

The text is clear and well-tuned to the targeted audience. The supplementary information is mostly clear, apart from the explanation of the scenarios.

References

References are mostly complete, relevant and traceable. A (selection of) reference(s) to the LCA method would be helpful.

Your expertise

I am not an expert in petrochemical refinery processes. Therefore, I cannot validate the inventory data for quality, consistency, and completeness.

Reviewer #3 (Remarks to the Author):

The authors comprehensively review and evaluate the CI in each country and estimate the implication of various jet fuel pathways on aviation emissions. The referenced numbers include sufficient references, and the research methodologies are sound.

Reviewer #4 (Remarks to the Author):

- What are the noteworthy results?

The analysis of global petroleum based well-to-wake fuel usage and GHG emissions, with improved insight by country-level and fuel type for variations in these components of the total, are noteworthy. The authors rightly assess that this serves as a useful baseline.

- Will the work be of significance to the field and related fields? How does it compare to the established literature? If the work is not original, please provide relevant references.

I believe the statement in lines 85-87, "To fill these gaps, we perform a first-of-its-kind LCA that evaluates the entire jet fuel life cycle (i.e., well-to-wake, which describes the life cycle GHG inventory of aviation fuels) GHG emissions." ...is not entirely accurate. Others have proposed and demonstrated methods for computing well-to-wake GHG emissions.

For example (though there are others):

- Agusdinata, D.B., Zhao, F., Ileyeji, K.E., DeLaurentis, D.A., "Life Cycle Assessment of Potential Biojet Fuel Production in the United States", *Environmental Science and Technology*, Vol. 45, No. 21, 2011, pp. 9133-9143.

DOI: 10.1021/es202148g

- Elgowainy, Amgad, et al. Life-cycle analysis of alternative aviation fuels in GREET. No. ANL/ESD/12-8. Argonne National Laboratory, 2012.

Perhaps this paper is first on global scale; if so this must be carefully qualified as such, while at the same time acknowledging prior work such as the above, but likely others, that have shown LCA well-to-wake modeling to include Sustainable Aviation Fuels.

- Does the work support the conclusions and claims, or is additional evidence needed?

My main concern is the depth of methodology, or lack thereof, in producing the results of Figure 4. The Supplemental Note 2 offers little explanation or justification. There have been far more detailed studies that address the potential fleetwide emissions benefits of aircraft and air traffic technology improvements. At a minimum, these should be referenced and used to justify reduction percentages used to create Fig. 4. For example:

- Moolchandani, K., Govindaraju, P., Roy, S., Crossley, W. A., and DeLaurentis D. A., "Assessing Effects of Aircraft and Fuel Technology Advancement on Select Aviation Environmental Impacts." AIAA Journal of Aircraft.

DOI: 10.2514/1.C033861

- Hassan, Mohammed, Holger Pfaender, and Dimitri Mavris. "Probabilistic assessment of aviation CO2 emission targets." Transportation Research Part D: Transport and Environment 63 (2018): 362-376.

- Are there any flaws in the data analysis, interpretation and conclusions? - Do these prohibit publication or require revision?

I would not say I identified flaws, just insufficient depth of modeling and evidence for the results in Figure 4, per the above. I did not attempt to run their code and reproduce results. Frankly, I would argue to eliminate Fig. 4 from the paper, it detracts from the main and noteworthy finding.

- Is the methodology sound? Does the work meet the expected standards in your field?

Yes

- Is there enough detail provided in the methods for the work to be reproduced?

I presume so.

Reviewer #1 (Remarks to the Author):

“This lack of self-sufficiency is due to refining capacity and market constraints”.

There are also countries where refining capacity is high but do not extract as much local crude oil as possible, e.g. Italy or Netherlands. This is due to local policies limiting local crude oil extraction. There can be various reasons. For example, Italy is limiting offshore areas where companies can explore for oil and gas in the Adriatic sea). The Netherlands is limiting extraction volumes in Groningen due to concerns related to local earthquakes).

“Turkey has the highest CI among major consumption countries mostly because the emissions associated with its crude oil production (13.2 gCO₂e MJ⁻¹ jet fuel) are ~50% higher than the global average production emissions”

Unclear why Turkey’s emissions are 50% higher than global average production emissions. Please add further information to allow the reader to understand this statement.

“With proper supply-chain decarbonization, up to 45% of the global jet fuel production can meet the eligibility of LCAF under CORSIA, and up to 2.3 GtCO₂e emissions can be avoided by 2050 ”

I would remove “up to 45% of the global jet fuel production can meet the eligibility of LCAF under CORSIA”. This depends on the standard value assumed under CORSIA, which will probably be updated between now and 2050 to reflect the decarbonization of all sectors including oil refining and supply chains.

“Having such a transparent jet fuel CI baseline can improve decision-making in each region by understanding the key emissions drivers and designing policy incentives (e.g., first replacing the most emissions-intensive petroleum jet fuel pathways with SAF) to achieve the ICAO carbon-neutral growth goal by 2050”.

I would remove “to achieve the ICAO carbon-neutral growth goal by 2050” since only mitigation can be supported by lower carbon intensity of petroleum-based jet fuels, not carbon neutrality.

“The high technology improvements scenario includes a 47% emissions reduction from the production and transportation of crude oil with no routine flaring and minimal fugitives & venting, carbon capture for all process units in refineries, and the use of low-carbon steam and electricity”

Please expand on the part “carbon capture for all process units in the refinery”. What emissions will be captured in the oil refineries, to what units we refer and what emissions we will not capture?

Extended Data Fig. 4: please round to first decimal (given uncertainties)

ADDITIONAL: I would add a reflection regarding the EU fossil fuel comparator, which is 94 gCO₂eq/MJ, which is a quite high value for jet fuels (as also can be well-argued in your article based on your findings). The main reason can probably be found in adopting marginal CO₂ refining emissions instead of energy allocation (you use in this article). Previous EU policies (renewable energy directive and fuel quality directives) were based on energy-allocation values and the fossil fuel comparator was 83.8.

Note:

- 1) all revisions are highlighted in yellow in the main manuscript and SI.**
- 2) all revisions are also copied here in blue to facilitate the reviewers.**
- 3) references cited below in blue sections can be found in the Main Manuscript.**

Reviewer #1's Comments

This study provides a noteworthy estimation of life cycle GHG emissions from petroleum jet fuel at the regional level.

The article is well-written, original and scientifically valuable and has a higher standard than the average articles published in this field.

I think that the significance of the findings in this field is high but probably not or just enough to be published in Nature Communications (also in light of the similar article recently published in Nature Climate change "Carbon intensity of global crude oil refining and mitigation potential").

The work with the respective methodology supports well the conclusions and claims.

The detail provided in the methods is enough transparent to allow reproduction.

***Response:** We would like to thank the reviewer for the encouraging comments above. We would also like to point out that the findings of this manuscript are novel and informative in understanding the life cycle GHG emissions of the global petroleum jet fuel supply chain.*

The paper we recently published in Nature Climate Change "Carbon intensity of global crude oil refining and mitigation potential" mainly focused on evaluating the carbon footprint of refining each barrel of crude oil as a whole (reported in kgCO_{2e}/bbl crude). It does not report refining emissions on a product level (e.g., gCO_{2e}/MJ jet fuel is not available).

The current manuscript under consideration advances our understanding of the life cycle GHG emissions of petroleum jet fuel that originate from the entire supply chain including crude oil extraction, transportation, and refining as well as jet fuel distribution and combustion. Although there have been studies in the literature tackling the life cycle GHG emissions of jet fuel, especially sustainable aviation fuel, our work here provides more insights into how the carbon footprint of petroleum jet fuel can be different across the globe, thanks to the geographically rich datasets we were able to access and the bottom-up engineering-based models (i.e., OPGEE, COPTM, and PRELIM) we used. Therefore, the results can serve as a high-resolution baseline to inform decisions about smarter use of sustainable aviation fuel (SAF) and other emissions reduction opportunities.

	Nature Climate Change paper (2020)	Current manuscript under consideration by Nature Communications
Scale	Global	Global
LCA functional unit	Per bbl crude oil	Per MJ petroleum jet fuel
Life cycle stages covered	Crude oil refining	Crude oil production, transportation, refining, jet fuel distribution, and combustion
Models used	PRELIM	OPGEE, COPTM, PRELIM

Reviewer #2's Comments

Key results

The authors conducted a detailed and thorough analysis of the well-to-wake greenhouse gas (GHG) footprint of global jet fuel consumption based on a process-based attributional life cycle assessment approach. They calculated a global-weighted average of 88.7 g CO₂eq MJ⁻¹ with ranges of 81.1 to 94.8 g CO₂eq MJ⁻¹ at the country level. Whilst emissions of combustion are constant, the variation upstream in the supply chain from crude oil production, transportation and refining provides important insights in the GHG footprint of the aviation sector and options and limitations to reduce it. To this purpose, the study also includes two future scenarios that explore GHG savings from technological development.

Response: We are grateful to the reviewer's positive and encouraging endorsement. In below, we have responded to the reviewer's comments with detailed responses.

Validity

A sensitivity check is included for key processes and results are correctly validated. However, there are a few points concerning the interpretation phase of LCA:

- Firstly, the limitations of the method and input data need some more elaboration. For example, the chosen allocation method of upstream emissions in the refinery (energy allocation) has a significant impact on the results; see Moretti et al. (2017).
- Secondly, the goal and scope of the study, applied method and results for the current well-to-wake GHG emissions are correctly aligned. However, the study also draws some conclusions and makes ambitious promises in the title (climate-neutral aviation) about the future (2050) that are inconsistent with the inventory data that is not representative for refineries operating in a climate-neutral energy sector. They will need to adjust their outputs to changing market demands.

Moretti, C., Moro, A., Edwards, R., Rocco, M.V. and Colombo, E., 2017. Analysis of standard and innovative methods for allocating upstream and refinery GHG emissions to oil products. *Applied Energy*, 206, pp.372-381.

Response:

Thank you for these comments. We are aware of the fact that the chosen allocation method (for both upstream and refining emissions in the refinery) can have considerable impacts on carbon intensity results. Based on this, we included a sensitivity analysis in the original manuscript to estimate the impact of different allocation methods on product-level GHG emissions including process-unit-level hydrogen-, mass-, and price-based as well as refinery-level energy, hydrogen-, mass-, and price-based methods. See Extended Data Fig. 5 and Supplementary Table 8 for more details. Unfortunately, we didn't have access to each refinery's LP model for marginal emissions calculations, therefore, the marginal method was

not tested. However, the results of the marginal emissions calculations in the Moretti et al. (2017) paper are very close to those of our model when the process-unit level allocation is employed (not shown in the Moretti et al. paper). The bigger difference is observed when refinery level allocation is employed, and we note this in our sensitivity analysis as well.

Moretti et al. (2017) is cited in our original manuscript (Ref. #14). Their paper argued that as compared to the marginal emissions method (which requires a full linear-programming model of any chosen refinery), two economic-value-based methods were recommended according to their consistency with performance criteria proposed by the authors.

Process-unit-level energy-based allocation was chosen as the baseline allocation method in our study because 1) it aligns with our previous studies and publications that adopted energy-based allocation in the LCA of energy systems (Masnadi et al. 2018 & 2021; Jing et al. 2020); 2) this study was intended to inform ICAO's LCA methods on lower carbon aviation fuels (LCAF), which advocates for energy-based allocation in its document to support Annex 16, Volume IV and is essential for the implementation of the CORSIA. See "CORSIA Methodology For Calculating Actual Life Cycle Emissions Values" published in June 2022, in which it says, quote, "Refinery models such as PRELIM may be used to estimate GHG emissions of crude refining to jet fuel. Jet-specific refinery CIs should be calculated using process-unit-level energy-based allocation among petroleum products when refinery models are used."; 3) it is a method that can be used by any stakeholder, regardless of their access to proprietary refinery LP models.

To accommodate the reviewer's comment on the allocation method, we have added the following statements to the revised manuscript:

Methods, PRELIM, bullet point 6: "The process-unit-level energy-based allocation method is chosen because it aligns well with recent studies on the life cycle emissions of energy systems in which the energy-based allocation was applied^{35,38,59}. Also note that the most recent CORSIA methods for calculating actual life cycle emissions values also adopted process-unit-level energy-based allocation method³⁹. All products (i.e., 16 in total) are included for allocation and a variety of allocation methods are tested in the sensitivity analysis."

Methods, Sensitivity Analysis: "Note that the choice of allocation (or displacement or marginal) method for co-product treatment can be arbitrary and therefore may have a substantial impact on the refining CI of jet fuel, leading to different results and interpretations. It can be misleading if mitigation options are compared by using a baseline that is not consistent in its methods as it can end up with negative unintended consequences. As such, GHG reduction policies and

strategies must be transparent about the methods and data used to calculate the baseline and ensure that any mitigation opportunity is based on consistent methods.”

SI Table 8, caption: “The marginal emissions associated with jet fuel refining are not calculated as a comparator because the refinery linear programming (LP) models are not available but are similar to those of the process-unit energy allocation results from Moretti et al. (2017)¹¹.”

We also agree that there needs to be more discussion on the limitations of the method and data. The following section has been added to the revised manuscript:

Policy Implications, last section: “It should be noted that the bottom-up attributional LCA approach adopted in this study provides an average baseline of jet fuel supply chain footprint. However, such an approach is not intended to be used to evaluate the indirect effects of changing markets and policies, which should be considered in future research. Other limitations and future work include: (1) regularly updating the latest crude oil production and transportation carbon intensity, (2) differentiating each individual refinery in PRELIM by matching the actual throughput and energy use of key process units, and (3) comparing the costs of decarbonization options of the petroleum supply chain with those of SAF to make economically sound decisions.”

Masnadi, M. S. et al. Global carbon intensity of crude oil production. Science (80). 361, 851–853 (2018)

Jing, L., El-Houjeiri, H.M., Monfort, J.C. et al. Carbon intensity of global crude oil refining and mitigation potential. Nat. Clim. Chang. 10, 526–532 (2020).

<https://doi.org/10.1038/s41558-020-0775-3>

Masnadi, M.S., Benini, G., El-Houjeiri, H.M. et al. Carbon implications of marginal oils from market-derived demand shocks. Nature 599, 80–84 (2021). <https://doi.org/10.1038/s41586-021-03932-2>

Secondly, we agree with the reviewer’s point on being somewhat ambitious in the title. Therefore, we have changed the title to

“Understanding the variability in life cycle emissions of petroleum jet fuel can lead to better decision-making on the way to decarbonizing aviation”

and have added more clarifications to the caption of Fig.4:

“The base case assumes a constant well-to-wake CI of 3.8 tonne CO_{2e} per tonne jet fuel (equivalent to 88.7 gCO_{2e} MJ⁻¹ jet fuel presented above for 2017) and a cumulative consumption of 11.5 Gt jet fuel over the period from 2019 to 2050, indicating that the cumulative emissions are 44 GtCO_{2e} (2019-2050). ”

“Other aspects not included in this forecasting: 1) advances in crude oil extraction and refinery operations that are more energy efficient and 2) shift in refinery product demand.”

Significance

The results are significant, but the title, conclusions, and recommendations must be consistent with the limitations of the method and input data.

Response: *We agree with the reviewer’s comment and have updated the title, conclusions, and recommendations:*

“Understanding the variability in life cycle emissions of petroleum jet fuel can lead to better decision-making on the way to decarbonizing aviation”

“Having such a transparent jet fuel CI baseline can improve decision-making amid the transition to decarbonizing aviation by understanding the key emissions drivers and designing policy incentives (e.g., first replacing the most emissions-intensive petroleum jet fuel pathways with SAF).”

Data and methodology

The authors provide a good description of the data sources and models used in the supplementary information and offer additional data upon request, which will be helpful to modelers and other practitioners who will use this manuscript. The method section has a few limitations that should be easy to address:

- The characterization of GHG emissions is not explained. If non-CO₂ GHG emissions are ignored, both the chosen scope and the implications to the results should be clearly explained in the main text.
- The explanation of the scenarios in the supplementary information is not very clear. A table with the scenario assumptions supported with argumentation would be helpful.

Response: *We fully agree with the reviewer that it is important to clarify if non-CO₂ GHG emissions are included in our study. On the crude oil production and transportation side, greenhouse gases included are CO₂, CO, CH₄, N₂O, and non-methane volatile organic compounds (NMVOCs) as described in the OPGEE manual; on the crude oil refining side,*

greenhouse gases included are CO₂, CH₄, N₂O, and non-methane volatile organic compounds (NMVOCs) as per described in the PRELIM manual; on the combustion side, according to ICAO's jet fuel conversion factor, only CO₂ is included. The following has been included in the revised manuscript (Methods section) to address the reviewer's concerns about non-CO₂ GHG emissions.

Methods, end of the Data sources and data processing section: "Note for crude oil production and transportation, greenhouse gases included are CO₂, CO, CH₄, N₂O, and non-methane volatile organic compounds (NMVOCs); for crude oil refining, greenhouse gases included are CO₂, CH₄, N₂O, and non-methane volatile organic compounds (NMVOCs); for jet fuel combustion, according to ICAO's jet fuel conversion factor, only CO₂ is included."

Analytical approach

No comments.

Clarity and context

The text is clear and well-tuned to the targeted audience. The supplementary information is mostly clear, apart from the explanation of the scenarios.

References

References are mostly complete, relevant and traceable. A (selection of) reference(s) to the LCA method would be helpful.

Your expertise

I am not an expert in petrochemical refinery processes. Therefore, I cannot validate the inventory data for quality, consistency, and completeness.

Response: We appreciate the reviewer's thorough comments and suggestions.

Suggested improvements

Comment 1: L1-2: the title is too ambitious regarding climate-neutral aviation and content covered in the manuscript. Optimizing fossil jet supply will not be sufficient to meet CORSIA objectives (climate-neutral growth).

Response: We thank the reviewer for shedding light on this matter. We have changed the title to something that is less ambitious but accurate enough to highlight our contributions:

"Understanding the variability in life cycle emissions of petroleum jet fuel can lead to better decision-making on the way to decarbonizing aviation"

Comment 2: L306: the section on policy limitations should also cover the limitations of attributional modeling concerning (policy) decision support and change-oriented scenario analyses.

Response: This is a good suggestion. Indeed, our bottom-up attributional approach captures the absolute supply chain footprint that can serve as a global average or baseline for evaluating emissions reductions opportunities. It is however not capable of dealing with the indirect effects of changing markets and policies, which highlights the need for future studies that develop and employ methods to capture market dynamics and how they might change over time. The following statement has been included in the revised manuscript to further discuss the limitations and future work:

Policy Implications, last section: “It should be noted that the bottom-up attributional LCA approach adopted in this study provides an average baseline of jet fuel supply chain footprint. However, such an approach is not intended to be used to evaluate the indirect effects of changing markets and policies, which should be considered in future research. Other limitations and future work include: (1) regularly updating the latest crude oil production and transportation carbon intensity, (2) differentiating each individual refinery in PRELIM by matching the actual throughput and energy use of key process units, and (3) comparing the costs of decarbonization options of the petroleum supply chain with those of SAF to make economically sound decisions.”

Comment 3: L213-214: a critical note on these investments, risks of lock-in in current refinery capacity, and the changing role of refineries in the energy transition would be helpful.

Response: We appreciate the reviewer’s comment. The following has been included in the revised manuscript to address the reviewer’s concern.

Well-to-Wake Life Cycle Carbon Intensity (CI) of Jet Fuel, last section: “However, it is worth noting that such potential is based on current crude oil refining throughput and may be subject to change of refinery operation modes (e.g., throughput change at refinery- and process unit-level and change of product slate to meet market demands) in the future.”

Comment 4: L316-327: it would be interesting to cover the challenges of the transition of the aviation sector in more detail. For example, liquid electrofuels (PtX) are not included in the text. Furthermore, significant amounts of biofuels will likely shift from road transport to aviation under updated support policies in the EU (RED II and RED III). See, for example, de Jong et al

(2018). de Jong, S., van Stralen, J., Londo, M., Hoefnagels, R., Faaij, A. and Junginger, M., 2018. Renewable jet fuel supply scenarios in the European Union in 2021–2030 in the context of proposed biofuel policy and competing biomass demand. *Gcb Bioenergy*, 10(9), pp.661-682.

Response: *We appreciate the reviewer’s comment. The types of SAF we included in the original manuscript and SI Table 9 are CORSIA eligible SAF recognized by ICAO. However, we agree with the reviewer that there are other pathways that can help decarbonize the aviation sector like liquid electrofuels (eSAF or power-to-liquid SAF). In addition, we also acknowledge the reviewer’s comment on how biofuels will likely shift quickly from road transport to aviation. The following section has been updated in the revised manuscript, the Policy Implications section:*

“In addition, biomass-based SAFs have raised questions about feedstock availability, land availability, conversion efficiency, and competition with other sectors (e.g., road transport and chemical market)⁴⁶. The spatial variation of jet fuel consumption and supply chain emissions among different regions further complicates this challenge and calls for a clearer understanding of how SAF can be best deployed in different regions to maximize emissions reductions. It has been recently reported that EU is likely to shift more biofuels from road transport to aviation⁴⁷⁻⁴⁹. Future work needs to assess other emerging technologies such as power-to-liquid SAF and hydrogen fuel cells^{50,51}.”

Comment 5: Finally, a recommendation for further research beyond the scope of the presented manuscript would be to assess marginal emissions of current and future jet fuel supply.

Response: *We are grateful to the reviewer for this comment. The following section has been included in the revised manuscript to reflect on the limitations and future work:*

Policy Implications, last section: “It should be noted that the bottom-up attributional LCA approach adopted in this study provides an average baseline of jet fuel supply chain footprint. However, such an approach is not intended to be used to evaluate the indirect effects of changing markets and policies, which should be considered in future research. Other limitations and future work include: (1) regularly updating the latest crude oil production and transportation carbon intensity, (2) differentiating each individual refinery in PRELIM by matching the actual throughput and energy use of key process units, and (3) comparing the costs of decarbonization options of the petroleum supply chain with those of SAF to make economically sound decisions.”

Reviewer #3's Comments

The authors comprehensively review and evaluate the CI in each country and estimate the implication of various jet fuel pathways on aviation emissions. The referenced numbers include sufficient references, and the research methodologies are sound.

Response: We appreciate the reviewer for acknowledging the importance of our work.

Reviewer #4's Comments

Comment 1: What are the noteworthy results?

The analysis of global petroleum based well-to-wake fuel usage and GHG emissions, with improved insight by country-level and fuel type for variations in these components of the total, are noteworthy. The authors rightly assess that this serves as a useful baseline.

Response: We appreciate the reviewer's acknowledgement of the importance of our work.

Comment 2: Will the work be of significance to the field and related fields? How does it compare to the established literature? If the work is not original, please provide relevant references.

I believe the statement in lines 85-87, "To fill these gaps, we perform a first-of-its-kind LCA that evaluates the entire jet fuel life cycle (i.e., well-to-wake, which describes the life cycle GHG inventory of aviation fuels) GHG emissions." ...is not entirely accurate. Others have proposed and demonstrated methods for computing well-to-wake GHG emissions.

For example (though there are others):

- Agusdinata, D.B., Zhao, F., Ileleji, K.E., DeLaurentis, D.A., "Life Cycle Assessment of Potential Biojet Fuel Production in the United States", Environmental Science and Technology, Vol. 45, No. 21, 2011, pp. 9133-9143.

DOI: 10.1021/es202148g

- Elgowainy, Amgad, et al. Life-cycle analysis of alternative aviation fuels in GREET. No. ANL/ESD/12-8. Argonne National Laboratory, 2012.

Perhaps this paper is first on global scale; if so this must be carefully qualified as such, while at the same time acknowledging prior work such as the above, but likely others, that have shown LCA well-to-wake modeling to include Sustainable Aviation Fuels.

*Response: We appreciate the reviewer's comment. Our work, as compared to the literature, is the first attempt to estimate the life cycle emissions of **petroleum jet fuel** (rather than sustainable aviation fuels) **on a global scale**. To acknowledge prior work and precisely position our paper in the literature, the following section has been updated including references suggested by the reviewer:*

"There have also been numerous life cycle carbon footprint studies on SAF¹²⁻¹⁶ but the understanding of LCAF has been limited due to the lack of geographically rich data (e.g., crude intake and jet fuel yield by refinery) and inconsistent assumptions (e.g., system boundary and allocation method) that result in a wide range of life cycle emissions estimates varying up to five-fold¹⁷⁻²⁶. To fill these

gaps, we perform a first-of-its-kind LCA that estimates the life cycle (i.e., well-to-wake, which describes the life cycle GHG inventory) emissions of petroleum jet fuel (rather than SAF) on a global scale”

Comment 3: Does the work support the conclusions and claims, or is additional evidence needed?

My main concern is the depth of methodology, or lack thereof, in producing the results of Figure 4. The Supplemental Note 2 offers little explanation or justification. There have been far more detailed studies that address the potential fleet wide emissions benefits of aircraft and air traffic technology improvements. At a minimum, these should be referenced and used to justify reduction percentages used to create Fig. 4. For example:

- Moolchandani, K., Govindaraju, P., Roy, S., Crossley, W. A., and DeLaurentis D. A., "Assessing Effects of Aircraft and Fuel Technology Advancement on Select Aviation Environmental Impacts." AIAA Journal of Aircraft.

DOI: 10.2514/1.C033861

- Hassan, Mohammed, Holger Pfaender, and Dimitri Mavris. "Probabilistic assessment of aviation CO₂ emission targets." Transportation Research Part D: Transport and Environment 63 (2018): 362-376.

Response: *We appreciate the reviewer’s feedback on such an important finding from our analysis. The caption of Fig.4 contains details about how we produced Fig.4. To address the concern raised by this reviewer, we have included more information in both Fig.4 caption and the revised SI - Supplemental Note 2.*

In addition, the estimates of GHG savings by 2050 we have for aircraft technology improvements and air traffic technology improvements are 36% and 13% (based on ICAO’s 2019 environment report), respectively, with the rest 51% savings come from SAF. The Hassan et al. (2018) paper reported something close to our estimates, such that aircraft technology improvements and air traffic technology improvements can save 31% and 5% of the cumulative GHG emissions in the U.S. by 2050, respectively, with the rest 64% savings come from SAF. Moolchandani et al. (2016) also estimated that aircraft and fuel technology advancement on a fleet level can save up to 50% of the CO₂ emissions from fuel use. Therefore, our method seems to provide reasonable estimates of potential GHG emissions savings. More references including those suggested by the reviewer have been included in the revised SI to acknowledge the fact that there have been numerous studies addressing the potential fleet wide emissions benefits associated with aircraft technology and ATM improvements.

Supplementary Note 2: “Fig. 4 in the main text first presents a projection of the cumulative well-to-wake baseline emissions for international aviation from 2019

to 2050, which are estimated to be 44 GtCO_{2e}. The following are the steps taken to obtain this estimate. The annual global jet fuel consumption data from 2005 to 2021 are obtained from the IEA World Oil Statistics²⁷, ranging from 233 to 333 million tonnes per year. The annual global jet fuel consumption in 2022, 2023, and 2024 are projected to be at the same level as 2015, 2018, and 2019 respectively²⁷. Jet fuel use from 2025 to 2050 is assumed to follow the growth in jet fuel demand projected by the ICAO²⁸ scenario of 1061 million tonnes per year between 2024 and 2050. The impact of fleet renewal and increasing fleet efficiency leading to reduced jet fuel consumption is considered in this projection. These estimates are made for domestic and international aviation combined. To investigate the impacts of international aviation alone, a fixed proportion of international to total jet fuel consumption of 57% (based on IEA 2017 data²⁷) is assumed for all years. That is, every year, 57% of the total global jet fuel use is assumed to be used for international aviation. By converting the well-to-wake GHG emissions of 88.7 gCO_{2e} MJ⁻¹ jet fuel presented in this study, we can get a life cycle emissions factor of 3.8 tonne CO_{2e} per tonne jet fuel ($88.7 \text{ gCO}_2\text{e/MJ} * 0.001 \text{ kg/g} * 42.8 \text{ MJ/kg jet fuel} = 3.8 \text{ tonne CO}_2\text{e/tonnes jet fuel}$). This emissions factor is then applied to get the well-to-wake GHG emissions (the thick solid line in Fig.4) from international aviation between 2005 and 2050. By summing up the numbers for every year between 2019 and 2050 (2019 is picked as the start of implementing emissions reductions), the cumulative jet fuel consumption and well-to-tank emissions from international aviation are 11.5 Gt jet fuel and 44 GtCO_{2e}, respectively. Note there are other mitigation opportunities that should be explored in future analyses as well shifts in demand that could affect these trajectories.

The dashed line represents ICAO's aspirational goal for international aviation of carbon neutral growth from 2019 levels, which is 727 GtCO_{2e} per year to 2050 (note that the original emissions baseline for CORSIA's carbon-neutral growth target is the average of 2019 and 2020, which was amended to the 2019 emission levels due to COVID-19)²⁹. The first two reduction categories are aircraft technology improvements and improved air traffic management (ATM) and infrastructure use, which have been in effect for some years. Aircraft technology improvements are aimed at propulsion, aerodynamics, weight reduction, and urban air mobility; ATM refers to safe and orderly traffic flow to avoid congestion and efficient airspace management for both civil and military use; and infrastructure use refers to operational improvement (e.g., ensuring the plane's engines are clean to developing and using new arrivals procedures). There have been several studies estimating potential fleet wide emissions benefits of aircraft and air traffic technology improvements in the literature^{30–33}. In this study,

according to the ICAO's 2019 environment report, we set the fuel use reductions from aircraft technology and operation (ATM and infrastructure use) improvements as 0.98% and 0.39% per annum²⁸. We then calculate the annual fuel use and GHG emissions savings (i.e., fuel use times 3.8 tonne CO_{2e} per tonne jet fuel as above) from both improvements by applying these exponentially decreasing rates for the period of 2019-2050, which are shown as the blue (7.8 GtCO_{2e}) and green areas (2.8 GtCO_{2e}) under the baseline curve in Fig. 4.

The potential GHG emissions savings from decarbonizing the jet fuel supply chain (see Methods, Jet fuel supply chain GHG emissions reduction technologies and scenarios, low and high technology improvements scenarios) can be 0.7 and 2.3 GtCO_{2e} by 2050. These savings only start from 2026 because that is when the predicted annual emissions less GHG savings from technology and infrastructure improvements are above the ICAO's carbon-neutral growth target. The GHG savings from the low and high technology improvements scenarios can be calculated by comparing their well-to-wake CI (i.e., 86.6 and 81.6 gCO_{2e} MJ⁻¹ jet fuel for low and high technology improvements, respectively) to the baseline CI of 88.7 gCO_{2e} MJ⁻¹ jet fuel and multiplying the CI differences with annual jet fuel use. Note that the low technology improvements scenario is a subset of the high technology improvements scenario.

The remaining emissions reductions required for the ICAO's carbon-neutral growth target (727 GtCO_{2e} per year) would be 8.8 GtCO_{2e} by 2050, if the high technology improvements are applied toward the jet fuel supply chain. This number would be 11.1 GtCO_{2e} if no supply chain decarbonisation is adopted. Such remaining reductions can be achieved via methods like carbon offsetting and low-carbon aviation fuels that meet specific sustainability criteria. Therefore, to keep every year's emissions at the ICAO's carbon-neutral growth target, a cumulative emission of 21.6 GtCO_{2e} needs to be reduced between 2019 and 2050.”

Fig.4 caption: “Only international aviation is considered. Jet fuel consumption data are from IEA (2005-2024)³³ and ICAO (2050)⁴. The original IEA data includes domestic and international consumption; therefore, a 57% share is calculated for the international portion based on the 2017 record and assumed for the whole period. The base case assumes a constant well-to-wake CI of 3.8 tonne CO_{2e} per tonne jet fuel (equivalent to 88.7 gCO_{2e} MJ⁻¹ jet fuel presented above for 2017) and a cumulative consumption of 11.5 Gt jet fuel over the period from 2019 to 2050, indicating that the cumulative emissions are 44 GtCO_{2e} (2019-

2050). Also, this base case estimate considers the impact of fleet renewal with increasing fleet efficiency and therefore reduced jet fuel consumption. Aircraft Technology improvements aim at propulsion, aerodynamics, weight reduction, and urban air mobility. ATM stands for air traffic management. Infrastructure use refers to operational improvement. These two (starting from 2019, regardless of jet fuel demand drop due to COVID-19) can improve fuel efficiency and reduce fuel consumption by 1.37% per annum (0.98% from technology and 0.39 from operations)⁴. Therefore, their impacts on emissions reductions from 2019 to 2050 are 7.8 and 2.7 GtCO_{2e}, respectively. The low and high technology improvement scenarios target carbon capture and low-carbon utilities, which are the same as explained in Fig. 2c and Methods. They can reduce 0.7 and 2.3 GtCO_{2e} by 2050 (starting from 2026 when the predicted annual emissions less technology and infrastructure improvements are above the ICAO's carbon-neutral growth target), respectively (the low technology improvements scenario is a subset of the high technology improvements scenario). The remaining emissions reductions required from SAF and carbon offsets are 8.8 GtCO_{2e} (11.1 GtCO_{2e} if no supply chain technology improvements are adopted). The carbon-neutral growth target line (based on the 2019 pre-pandemic level) requires 21.6 GtCO_{2e} emissions reductions by 2050. Note the original emissions baseline for CORSIA's carbon-neutral growth target is the average of 2019 and 2020, which was amended to the 2019 emission levels due to COVID-19⁵². See more details in Supplementary Note 2.”

Comment 4: Are there any flaws in the data analysis, interpretation and conclusions? - Do these prohibit publication or require revision?

I would not say I identified flaws, just insufficient depth of modeling and evidence for the results in Figure 4, per the above. I did not attempt to run their code and reproduce results. Frankly, I would argue to eliminate Fig. 4 from the paper, it detracts from the main and noteworthy finding.

Response: *We thank the reviewer for this comment. We feel that Fig.4 is a key element showing how important the supply chain decarbonisation is and how much SAF might be required to achieve the ICAO's carbon neutral growth target. Therefore, we would like to keep it in the main text. To accommodate the reviewer's concern on the insufficient depth of modeling and evidence for the results, we have added more details in the SI, Supplementary Note 2 to describe how this figure is plotted. See the response to Comment 3 above.*

Comment 5: Is the methodology sound? Does the work meet the expected standards in your field? Yes.

Comment 6: Is there enough detail provided in the methods for the work to be reproduced? I presume so.

Response: *We appreciate the reviewer for acknowledging the quality and soundness of our work.*

**Additional Comments from the attachment
(named “1_reviewer_attachment_1_1655831116”)**

Comment 1: “This lack of self-sufficiency is due to refining capacity and market constraints”. There are also countries where refining capacity is high but do not extract as much local crude oil as possible, e.g. Italy or Netherlands. This is due to local policies limiting local crude oil extraction. There can be various reasons. For example, Italy is limiting offshore areas where companies can explore for oil and gas in the Adriatic Sea). The Netherlands is limiting extraction volumes in Groningen due to concerns related to local earthquakes).

Response: We agree the reviewer’s comment that there are many reasons why a country would choose not to extract as much local crude oil as possible. However, here the self-sufficiency refers to jet fuel rather than crude oil. There are some countries (e.g., U.K., Germany, and Italy) that choose to import most of their jet fuel from other regions (e.g., the Middle East, Netherlands, and Korea) rather than satisfying their domestic jet fuel demand. This might be attributed to several factors such as refinery capacity constraints (not enough refineries producing jet fuel), market demand and pricing for refined products (producing more gasoline and diesel rather than jet fuel because of price differentials), and the price differential between domestic jet fuel and imported jet fuel. The following sentence has been included in the revised manuscript:

Global Jet Fuel Market, last paragraph: “This lack of self-sufficiency in jet fuel may be due to refining capacity constraints, market demand and pricing for refined products, and jet fuel price differential across the globe.”

Comment 2: “Turkey has the highest CI among major consumption countries mostly because the emissions associated with its crude oil production (13.2 gCO_{2e} MJ⁻¹ jet fuel) are ~50% higher than the global average production emissions”. Unclear why Turkey’s emissions are 50% higher than global average production emissions. Please add further information to allow the reader to understand this statement.

Response: Jet fuel CI is estimated by estimating the emissions associated with the refinery operations in a particular refinery and adding the crude oil production emissions for all crudes processed by that refinery. For example, if a refinery processes crudes A, B, and C, then the crude oil production CI of jet fuel is the volume weighted average of the production CI of crudes A, B, and C (by applying a conversion factor MJ crude/MJ jet fuel which is normally close to one) plus the refinery emissions assigned to process crude oil into jet fuel. Country-level results (e.g., 13.2 gCO_{2e} MJ⁻¹ jet fuel for Turkey as a volume weighted average value) can be derived based on the jet fuel production volume of each refinery within that Country as well as the crude oil production CI of that each refinery uses to produce jet fuel. Therefore, the main reason for Turkey-produced jet fuel having such a high crude oil

production CI is because Turkish refineries are heavily dominated by heavy crude oil imported from Iran and Iraq (70% of the crude oil refined in Turkey is from Iran and Iraq, the remaining 30% is from Saudi Arabia, Kuwait, and Russia that have lower crude oil production CI than the global average) that has remarkably higher production CI (13-21 gCO_{2e} MJ⁻¹ jet fuel) than the global average (8.6 gCO_{2e} MJ⁻¹ jet fuel). To facilitate the reader's understanding, the following section has been included in the revised manuscript:

Right after Fig. 2: “Turkey has the highest CI among major jet fuel consuming countries because 97% of its jet fuel consumption is from domestic refineries, which are heavily dominated by crude oil imported from Iran and Iraq that has higher crude oil production CI (12.9-21.4 gCO_{2e} MJ⁻¹ jet fuel) than the global average.”

Comment 3: “With proper supply-chain decarbonization, up to 45% of the global jet fuel production can meet the eligibility of LCAF under CORSIA, and up to 2.3 GtCO_{2e} emissions can be avoided by 2050”. I would remove “up to 45% of the global jet fuel production can meet the eligibility of LCAF under CORSIA”. This depends on the standard value assumed under CORSIA, which will probably be updated between now and 2050 to reflect the decarbonization of all sectors including oil refining and supply chains.

***Response:** This comment is helpful and constructive to improving the quality of our study. We agree with the reviewer that the CORSIA eligible fuels standard might change in the future and therefore the amount of jet fuel that can meet future standards would be unknown and uncertain. We have revised the wording in the revised manuscript:*

End of the Policy Implications section: “With proper supply-chain decarbonisation, up to 2.3 GtCO_{2e} emissions can be avoided by 2050.”

Comment 4: “Having such a transparent jet fuel CI baseline can improve decision-making in each region by understanding the key emissions drivers and designing policy incentives (e.g., first replacing the most emissions-intensive petroleum jet fuel pathways with SAF) to achieve the ICAO carbon-neutral growth goal by 2050”. I would remove “to achieve the ICAO carbon-neutral growth goal by 2050” since only mitigation can be supported by lower carbon intensity of petroleum-based jet fuels, not carbon neutrality.

***Response:** We agree with the reviewer on this. The carbon neutrality part has been removed from the revised manuscript:*

End of the Policy Implications section: “Having such a transparent jet fuel CI baseline can improve decision-making amid the transition to decarbonizing

aviation by understanding the key emissions drivers and designing policy incentives (e.g., prioritizing the replacement of the most emissions-intensive petroleum jet fuel pathways with SAF).”

Comment 5: “The high technology improvements scenario includes a 47% emissions reduction from the production and transportation of crude oil with no routine flaring and minimal fugitives & venting, carbon capture for all process units in refineries, and the use of low-carbon steam and electricity”. Please expand on the part “carbon capture for all process units in the refinery”. What emissions will be captured in the oil refineries, to what units we refer and what emissions we will not capture?

***Response:** We appreciate the reviewer for making this constructive comment. The refinery emissions reduction scenarios are adopted from our previous study (Jing et al. 2020), including post-combustion carbon capture (CC) applied to individual refinery process units and refinery-wide low-carbon utilities substitution. The process-unit modeling framework enables PRELIM to calculate emissions reductions from selected process units.*

For the refining stage under the low technology improvements scenario, we assume that post-combustion CC is ONLY applied to fluid catalytic cracking unit (FCC) and steam methane reformer (SMR) for all refineries. This is because these two are usually the most emissions intensive units across any given refinery and it is technically feasible to install CC equipment under most circumstances. FCC in general converts heavy oil fractions to value-adding light products that can be either blended as gasoline and diesel or processed for producing LPG and petrochemical feedstock. The most important emissions source of FCC is the carbonaceous deposit (coke) on the surface of the catalyst that needs to be burnt off regularly. On the other hand, SMR produces hydrogen from natural gas and high temperature steam (also produces CO₂ as a by-product). It is the main hydrogen source that provides hydrogen for hydroprocessing and is usually responsible for 30%-50% of the GHG emissions from a refinery. The reaction process is endothermic, which means there needs to be a heat source. In PRELIM, we assume natural gas and refinery fuel gas are combusted to provide such heat requirement. Therefore, CO₂ from both combustion and chemical conversion of natural gas and refinery fuel gas are available for CC.

For the refining stage under the high technology improvements scenario, in addition to CC being applied to FCC and SMR, we assume CC is also applied to capture and sequester CO₂ from natural gas combustion for providing heat to key process units including atmospheric tower, vacuum tower, coker, residue hydrocracker, gasoil hydrocracker, all hydrotreaters, naphtha reformer, desalter, kerosene merox unit, alkylation unit, isomerization unit, and fuel gas treatment unit. In addition, low-carbon electricity and steam are adopted for crude oil refining. Low-carbon electricity is available from sources such as nuclear- or renewable-

powered electricity generating facilities, while lower-carbon steam may be generated from CHP systems or renewable driven heat-generating facilities (e.g., solar steam). Emissions not captured (but counted as refining emissions) are indirect (Scope 2) emissions associated with electricity and natural gas production, and support services emissions (flaring and fugitive emissions).

We assume an amine-based post-combustion CC with an 85% capture efficiency as suggested by literature recommendations (de Mello et al., 2009; Shakerian et al., 2015; Stec et al., 2015). Hot flue gas captured from combustion processes passes through a waste-heat steam generator first, followed by CO₂ absorption in aqueous monoethanolamine. The absorption medium can be regenerated via heat. To meet water and pressure specifications, CO₂ compression and dehydration are also required. Energy (5.97 MJ steam and 0.2 kWh electricity per kg capture CO₂ as suggested by Motazedi et al., 2017) required by the CC equipment is assumed to be provided by low-carbon electricity and steam as described above.

For emissions reductions associated with crude oil production and transportation, based on literature recommendations (Masnadi et al., 2018), we assume a 16.7% emissions reduction through routine gas flaring minimization for the low technology improvement scenario. We assume a 47% emissions reduction through no routine gas flaring and minimal fugitives & venting for the high technology improvement scenario.

Jing, L. et al. Carbon intensity of global crude oil refining and mitigation potential. *Nat. Clim. Chang.* (2020) doi:10.1038/s41558-020-0775-3.

de Mello, L. F. et al. A technical and economical evaluation of CO₂ capture from FCC units. *Energy Procedia* 1, 117–124 (2009).

Shakerian, F., Kim, K. H., Szulejko, J. E. & Park, J. W. A comparative review between amines and ammonia as sorptive media for post-combustion CO₂ capture. *Appl. Energy* 148, 10–22 (2015).

Stec, M. et al. Pilot plant results for advanced CO₂ capture process using amine scrubbing at the Jaworzno II power plant in Poland. *Fuel* 151, 50–56 (2015).

Motazedi, K., Abella, J. P. & Bergerson, J. A. Techno-economic evaluation of technologies to mitigate greenhouse gas emissions at North American refineries. *Environ. Sci. Technol.* 51, 1918–1928 (2017).

Masnadi, M. S. et al. Global carbon intensity of crude oil production. *Science* 361, 851–853 (2018).

To respond to the reviewer's comment and to facilitate the readers, the following details are included in the revised Methods section, Jet fuel supply chain GHG emissions reduction technologies and scenarios:

“In this study, for the refining stage under the low technology improvements scenario, we assume that post-combustion CC is ONLY applied to fluid catalytic cracking unit (FCC) and steam methane reformer (SMR) for all refineries. This is because these two are usually the most emissions intensive units across any given refinery and it is technically feasible to install CC equipment under most circumstances. FCC in general converts heavy oil fractions to value-adding light products that can be either blended as gasoline and diesel or processed for producing LPG and petrochemical feedstock. The most important emissions source of FCC is the carbonaceous deposit (coke) on the surface of the catalyst that needs to be burnt off regularly. On the other hand, SMR produces hydrogen from natural gas and high temperature steam (also produces CO₂ as a by-product). It is the main hydrogen source that provides hydrogen for hydroprocessing and is usually responsible for 30%-50% of the GHG emissions from a refinery. The reaction process is endothermic, which means there needs to be a heat source. In PRELIM, we assume natural gas and refinery fuel gas are combusted to provide such heat requirement. Therefore, CO₂ from both combustion and chemical conversion of natural gas and refinery fuel gas are available for CC.

For the refining stage under the high technology improvements scenario, in addition to CC being applied to FCC and SMR, we assume CC is also applied to capture and sequester CO₂ from natural gas combustion for providing heat to key process units including atmospheric tower, vacuum tower, coker, residue hydrocracker, gasoil hydrocracker, all hydrotreaters, naphtha reformer, desalter, kerosene merox unit, alkylation unit, isomerization unit, and fuel gas treatment unit. In addition, low-carbon electricity and steam are adopted for crude oil refining. Low-carbon electricity is available from resources such as nuclear- or renewable-powered electricity generating facilities, while lower-carbon steam may be generated from CHP systems or renewable driven heat-generating facilities. Emissions not captured (but counted as refining emissions) are indirect (Scope 2) emissions associated with electricity and natural gas production, and support services emissions (flaring and fugitive emissions).

We assume an amine-based post-combustion CC with an 85% capture efficiency as suggested by literature recommendations⁶⁰⁻⁶². Hot flue gas captured from combustion processes passes through a waste-heat steam generator first, followed by CO₂ absorption in aqueous monoethanolamine. The absorption medium can be regenerated via heat. To meet water and pressure specifications, CO₂ compression and dehydration are also required. Energy (5.97 MJ steam and 0.2 kWh electricity per kg capture CO₂ as suggested by Motazedi et al., 2017⁶³)

required by the CC equipment is assumed to be provided by low-carbon electricity and steam as described above.

For emissions reductions associated with crude oil production and transportation, based on literature recommendations³⁸, we assume a 16.7% emissions reduction through routine gas flaring minimization for the low technology improvement scenario. We assume a 47% emissions reduction through no routine gas flaring and minimal fugitives & venting for the high technology improvement scenario.”

Comment 6: Extended Data Fig. 4: please round to first decimal (given uncertainties).

Response: Numbers in Extended Data Fig.4 have been rounded to first decimal.

Comment 7: ADDITIONAL: I would add a reflection regarding the EU fossil fuel comparator, which is 94 gCO₂eq/MJ, which is a quite high value for jet fuels (as also can be well-argued in your article based on your findings). The main reason can probably be found in adopting marginal CO₂ refining emissions instead of energy allocation (you use in this article). Previous EU policies (renewable energy directive and fuel quality directives) were based on energy-allocation values and the fossil fuel comparator was 83.8.

Response: We appreciate the reviewer’s comment and suggestion. The EU fossil fuel comparator has recently changed from 83.8 to 94.0 gCO₂eq/MJ for the evaluation of biofuels on a life cycle basis and it would make a good comparison to our results. A review of the literature shows that the comparator values for gasoline, diesel, and heavy fuel oil are 93.3, 95.1, and 94.2 gCO₂eq/MJ, respectively. Despite not having a dedicated comparator for jet fuel, our analysis shows that EU’s petroleum jet fuel life cycle emissions intensity is 89.1 gCO₂eq/MJ, which is fairly close to the latest EU fossil fuel comparator. The following sentence has been added to the revised manuscript:

*Well-to-Wake Life Cycle Carbon Intensity (CI) of Jet Fuel section, paragraph 2:
“The EU’s average obtained from this study is 89.1 gCO₂e MJ⁻¹, which is comparable to the EU fossil fuel comparator (94.0 gCO₂e MJ⁻¹, based on the marginal method, see Methods for more details)⁴¹.”*

In addition, we would like to argue that any allocation/displacement method selected for LCA studies can be arbitrary and therefore may lead to different results and interpretations. It can be misleading if we start comparing mitigation options using a baseline that is not consistent in its methods because it could end up with negative unintended consequences. As such, policies such as Fuel Quality Directive, ReFuelEU Aviation Initiative, and FuelEU Maritime Initiative must be transparent about the methods and data used to calculate the baseline and ensure that any mitigation opportunity uses consistent methods.

Methods, Sensitivity Analysis: “Note that the choice of allocation (or displacement or marginal) method for co-product treatment can be arbitrary and therefore may have a substantial impact on the refining CI of jet fuel, leading to different results and interpretations. It can be misleading if mitigation options are compared by using a baseline that is not consistent in its methods as it can end up with negative unintended consequences. As such, GHG reduction policies and strategies must be transparent about the methods and data used to calculate the baseline and ensure that any mitigation opportunity is based on consistent methods.”